# Perceptual adjustment queries and an inverted measurement paradigm for low-rank metric learning

**Austin Xu**[*]
Georgia Institute of Technology

**Andrew D. McRae**
EPFL

**Jingyan Wang**
Georgia Institute of Technology

**Mark A. Davenport**
Georgia Institute of Technology

**Ashwin Pananjady**
Georgia Institute of Technology

## Abstract

We introduce a new type of query mechanism for collecting human feedback, called the perceptual adjustment query (PAQ). Being both informative and cognitively lightweight, the PAQ adopts an inverted measurement scheme, and combines advantages from both cardinal and ordinal queries. We showcase the PAQ in the metric learning problem, where we collect PAQ measurements to learn an unknown Mahalanobis distance. This gives rise to a high-dimensional, low-rank matrix estimation problem to which standard matrix estimators cannot be applied. Consequently, we develop a two-stage estimator for metric learning from PAQs, and provide sample complexity guarantees for this estimator. We present numerical simulations demonstrating the performance of the estimator and its notable properties.

## 1   Introduction

Should we query cardinal or ordinal data from people? This question arises in a broad range of applications, such as in conducting surveys [1–3], grading assignments [4, 5], evaluating employees [6], and comparing or rating products [7, 8], to name a few. *Cardinal* data are numerical scores. For example, teachers score writing assignments in the range of 0-100, and survey respondents express their agreement with a statement on a scale of 1 to 7. *Ordinal* data are relations between items, such as pairwise comparisons (choosing the better item in a pair) and rankings (ordering all or a subset of items). There is no free lunch, and both cardinal and ordinal queries have pros and cons.

On the one hand, collecting ordinal data is typically more efficient in terms of worker time and cognitive load [9], and surprisingly often matches or exceeds the accuracy of cardinal data [1, 9]. The information contained in ordinal queries, however, is fundamentally limited and lacks expressiveness. For example, pairwise comparisons elicit binary responses where two items are compared against each other, but the absolute placement of these items with respect to the entire pool is lost. On the other hand, cardinal data are more expressive [10]. For example, assigning two items scores of 1 and 2 conveys a very different message from assigning them scores of 9 and 10, or 1 and 10, although all yield the same pairwise comparison outcome. However, the expressiveness of cardinal data often comes at the cost of miscalibration: Prior work has shown that different people have different scales [11], and even a single person's scale can drift over time (e.g., [12, 13]). These inter-person and intra-person discrepancies make it challenging to interpret and aggregate raw scores effectively.

The goal of this paper is to study whether one can combine the advantages of cardinal and ordinal queries to achieve the best of both worlds. Specifically, we pose the research question:

---

[*]Contact: `axu@gatech.edu`. The full version of this work can be found on arXiv.

37th Conference on Neural Information Processing Systems (NeurIPS 2023).

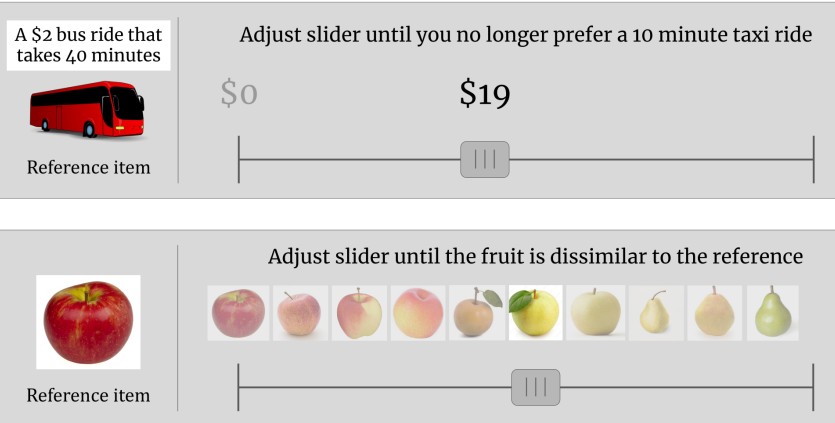

Figure 1: The user interface for perceptual adjustment query (PAQ) for preference learning (top) and similarity learning (bottom).

> *Can we develop a new paradigm for human data elicitation that is expressive, accurate, and cognitively lightweight?*

Towards this goal, we extract key features of both cardinal and ordinal queries, and we propose a new type of query scheme that we term the *perceptual adjustment query* (PAQ). As a thought experiment, consider the task of learning an individual's preferences between modes of transport. The query can take the following forms:

- **Ordinal:** Do you prefer a \$2 bus ride that takes 40 minutes or a \$25 taxi that takes 10 minutes?
- **Cardinal:** On a scale of 0 to 1, how much do you value a \$2 bus ride that takes 40 minutes?
- **Proposed approach:** To reach the same level of preference for a \$2 bus trip that takes 40 minutes, a taxi that takes 10 minutes would cost \$$x$.

A user interface for the proposed approach is shown in Figure 1 (top). We present the user a reference item (a \$2 bus ride that takes 40 minutes) and a sliding bar representing the number of dollars ($x$) for the 10-minute taxi cost. As the user adjusts the slider, the value of $x$ starts with 0 and gradually increases on a continuous scale. The user is instructed to place the slider at a point where they equally prefer a \$2 bus ride and a taxi ride of $x$ dollars.[2] The PAQ thus combines ordinal and cardinal elicitation in an intuitive fashion: We obtain ordinal information by asking the user to make cognitive judgments in a relative sense by comparing items, and cardinal information can be extracted from the location of the slider. The ordinal reasoning endows the query with accuracy and efficiency, while the cardinal output enables a more expressive response. Moreover, this cardinal output mitigates miscalibration, because instead of asking the user to rate on a subjective and ambiguous notion (i.e., preference), we provide the user a reference object (i.e., the \$2 bus ride) to anchor their rating scale.

Beyond combining the strengths of cardinal and ordinal queries, PAQs have additional advantages that are well illustrated with the example in Figure 1 (bottom). First, PAQs provide users with the *context* of a specific (continuous) dimension along which items vary. For example, consider a pairwise comparison between the reference item and the "yellow apple" selected in Figure 1. They have similar shapes, but different colors. If these two items are shown to the user in isolation, the user lacks context to judge whether they should be considered similar or dissimilar. In contrast, the full spectrum provided in PAQs tells the user that the similarity judgment is apples vs. pears. The access to such context improves self-consistency in user responses [14]. Second, PAQs provide "hard examples" by design and thus enable effective learning. Consider Figure 1 (bottom): Items on the left of the spectrum are apples (clearly similar to the reference), and items on the right are pears (clearly dissimilar to the reference), and only a small subset of items in the middle appear ambiguous. PAQs collect information precisely about "confusing" items in this ambiguous region. On the other hand, if ordinal queries are constructed by selecting uniformly at random from the items shown, an item in the ambiguous region will rarely be presented to the user.

---

[2]The ordinal component is crucial in our proposed perceptual adjustment query— we provide a reference item and instruct people to make a relational judgment of the target item compared to the reference item. Hence, the perceptual adjustment query is distinct from sliding survey questions that elicit purely cardinal responses.

The advantages of PAQs makes their deployment appealing in a variety of problem settings. For example, practitioners may deploy PAQs to learn human preferences, like in the taxi and bus example in Figure 1 or more complex settings such as housing preferences, where multiple features (price, square footage, proximity to employment etc.) vary as the slider is moved. PAQs can also be used to learn models for human perception, such as characterizing the extent of color blindness for a given user. One can present a user with red-green color blindness a sequence of colors that slowly transitions between red and green, and ask them to drag the slider until they perceive a difference in colors. An extended discussion of applications is provided in the full version of this paper on arXiv. In this paper, we focus on learning metrics to characterize human perception. In this problem, items are represented by points in a (possibly high-dimensional) space, and the goal is to learn a distance metric such that a smaller distance between a pair of items means that they are semantically and perceptually closer, and vice versa. Figure 1 (bottom) presents a PAQ for collecting similarity data for metric learning, where the user is instructed to place the slider at the precise point where the object appears to transition from being similar to dissimilar. To construct a sequence of images as shown in Figure 1 (bottom), one can traverse a path in the latent space of a generative model — given a latent feature vector, the generative model synthesizes a corresponding image.

## 1.1 Do PAQs improve upon ordinal queries? A simulation vignette

Consider the problem of Mahalanobis metric learning, which forms the focus of this paper. In this setting, items are represented as points in the vector space $\mathbb{R}^d$, which is in turn endowed with a Mahalanobis metric parametrized by a symmetric positive semidefinite matrix $\mathbf{\Sigma}^\star \in \mathbb{R}^{d \times d}$. The (dis-)similarity of two items is determined by their distance under the metric: The larger the (squared) distance $\|x - x'\|^2_{\mathbf{\Sigma}^\star} = (x - x')^\top \mathbf{\Sigma}^\star (x - x')$ between two items $x$ and $x'$ is, the more dissimilar the items are. We are particularly interested in the setting in which $\mathbf{\Sigma}^\star$ is *low-rank*. Established approaches in metric learning use ordinal queries, such as pairwise comparisons ("Are items $x$ and $x'$ similar?") [15–18], triplet comparisons [19] ("Which of the two items $x_1$ and $x_2$ is closer to reference item $x_0$?"), and ranking$-k$ queries ("Given a reference item $x_0$, rank the set of items $x_1, \ldots, x_k$ in terms of similarity to $x_0$") [14].

We compare the performance of such queries against PAQs in a toy metric learning setup. In particular, we choose a random low-rank matrix $\mathbf{\Sigma}^\star$ in dimension 50 with rank 10 (see Appendix B for our precise construction) and use the models and state-of-the-art algorithms of [19, 14] to produce pairwise, triplet, and ranking-$k$ queries and estimate the low-rank metric. In addition to these ordinal queries, we simulate PAQ responses under the model presented in Section 2 and use our algorithm (see Section 3) for estimation. To simplify the example, all query responses are generated in a *noiseless* fashion—for example, the triplet comparison always returns the closer item to the reference. We present our results in Figure 2, which illustrates a significant gap in information richness between PAQs and a variety of ordinal queries. The number of PAQ responses needed to attain a reasonable normalized error is dramatically lower than those of typical ordinal queries, illustrating that PAQs can greatly improve upon the performance of existing ordinal queries for metric learning. The rest of our paper explores this opportunity: It aims to make the deployment of PAQs theoretically grounded by designing provable methodology for learning a low-rank metric from PAQ responses.

## 1.2 Our contributions and organization

In addition to introducing the *perceptual adjustment query* (PAQ), we demonstrate its applicability to metric learning under a Mahalanobis metric. We first present a mathematical formulation of this estimation problem in Section 2. We then show that the sliding bar response can be viewed as an *inverted measurement* of the metric matrix that we want to estimate, which allows us to restate our problem as that of estimating a low-rank matrix from a specific type of trace measurement (Section 3). However, our PAQ formulation differs from classical matrix estimation due to two technical challenges: (a) the sensing matrices and noise are correlated, and (b) the sensing matrices are heavy-tailed. As a result, standard matrix estimation algorithms give rise to *biased estimators*. We propose a query procedure and an estimator that overcome these two challenges, and we prove statistical error bounds on the estimation error (Section 4). The unconventional nature of the sensing model and estimator causes unexpected behaviors in our error bounds; in Section 5, we present simulations verifying that these behaviors also appear in practice.

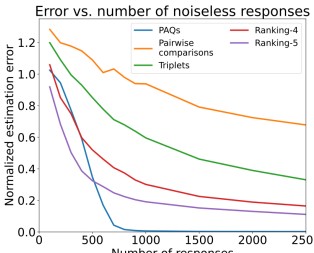

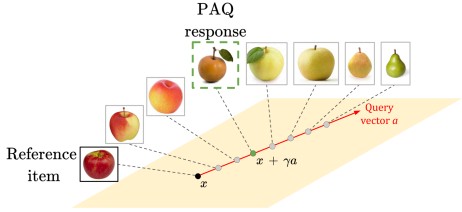

Figure 2: Performance of noiseless PAQs and various ordinal queries for low-rank metric learning. We plot the mean and standard error of the mean (shaded regions, not visible) of the normalized estimation error over 10 trials.

Figure 3: The perceptual adjustment query. Given a reference item $\boldsymbol{x}$ and a query vector $\boldsymbol{a}$, a continuous path of items is formed $\{\boldsymbol{x} + \gamma\boldsymbol{a} : \gamma \in [0, \infty)\}$. Then, a user is asked to pick the first item along this path that is dissimilar to the reference item, denoted by $\boldsymbol{x} + \gamma\boldsymbol{a}$.

## 1.3 Related work

We discuss related work on metric learning and the statistical techniques that we use.

**Metric learning.** As discussed in Section 1.1, prior work in metric learning [20] considers a wide variety of ordinal queries. PAQ can be viewed as extending a discrete set of items presented to users to a continuous spectrum, which is natural when one uses a generative model such as a GAN [21, 22]. However, the goal of tuple queries is to rank the items, whereas in PAQ the ranking is provided by the feature space and we ask people to identify a transition point (similar vs. dissimilar) in this ranking.

**Statistical techniques.** In our theoretical results, we apply techniques from the high-dimensional statistics literature. Our theoretical formulation (presented in Section 3) resembles the problem of low-rank matrix estimation from trace measurements (e.g., [23–28]; see [29] for a more complete overview), and, in particular, when the sensing matrix is of rank one [30–33]. However, as discussed in Section 3, our model presents two important distinctions from prior literature. In our case, the sensing matrices are both heavy-tailed and correlated with the measurement noise. The heavy-tailed matrices violate the assumptions of much prior work that relies on sub-Gaussian or sub-exponential assumptions on the sensing matrices. Prior work has attempted to address the challenge of heavy tails with methods such as robust loss functions [34, 35] or the "median-of-means" approach [36–38]. We draw particular inspiration from Fan et al. [39], which considers truncation to control heavy-tailed behavior for a variety of problems. However, in the low-rank matrix estimation setting, Fan et al. [39] only analyze the case of heavy-tailed noise under a sub-Gaussian design, meaning their methodology and results are not applicable to our problem setting.

## 2 Model

In this section, we present our model for the perceptual adjustment query (PAQ) in the context of its application to metric learning.

### 2.1 Mahalanobis metric learning

We consider a $d$-dimensional feature space where each item is represented by a point in $\mathbb{R}^d$. The distance metric model for human similarity perception posits that there is a metric on $\mathbb{R}^d$ that measures how dissimilar items are perceived to be. A recent line of work [40, 41] has modeled the distance metric as a Mahalanobis metric. If $\boldsymbol{\Sigma}^\star \in \mathbb{R}^{d \times d}$ is a symmetric positive semidefinite (PSD) matrix, the squared Mahalanobis distance with respect to $\boldsymbol{\Sigma}^\star$ between items $\boldsymbol{x}$ and $\boldsymbol{x}' \in \mathbb{R}^d$ is $\|\boldsymbol{x} - \boldsymbol{x}'\|_{\boldsymbol{\Sigma}^\star}^2 := (\boldsymbol{x} - \boldsymbol{x}')^\top \boldsymbol{\Sigma}^\star (\boldsymbol{x} - \boldsymbol{x}')$. The distance represents the extent of dissimilarity between items $\boldsymbol{x}$ and $\boldsymbol{x}'$: If we further have a perceptual boundary value $y > 0$, this model posits that items $\boldsymbol{x}, \boldsymbol{x}'$ are perceived as similar if $\|\boldsymbol{x} - \boldsymbol{x}'\|_{\boldsymbol{\Sigma}^\star}^2 < y$ and dissimilar if $\|\boldsymbol{x} - \boldsymbol{x}'\|_{\boldsymbol{\Sigma}^\star}^2 \geq y$. We adopt a high-dimensional framework and, following [19, 41], assume that the matrix $\boldsymbol{\Sigma}^\star$ is low-rank.

Note that if the goal is to predict whether two items are similar or dissimilar via computing the relation $\|\boldsymbol{x} - \boldsymbol{x}'\|_{\boldsymbol{\Sigma}^\star}^2 \gtrless y$, then this problem is scale-invariant, in the sense that two items are predicted as similar (or dissimilar) according to $(\boldsymbol{\Sigma}^\star, y)$, if and only if they are predicted as similar (or dissimilar)

according to $(c_{\mathsf{scale}}\boldsymbol{\Sigma}^\star, c_{\mathsf{scale}}y)$ for any scaling factor $c_{\mathsf{scale}} > 0$. We are thus interested in finding the equivalence class of solutions $\{(c_{\mathsf{scale}}\boldsymbol{\Sigma}^\star, c_{\mathsf{scale}}y) : c_{\mathsf{scale}} > 0\}$. Therefore, in practice, one can set $y$ to be any positive scalar and then estimate the corresponding $\boldsymbol{\Sigma}^\star$. Indeed, our theoretical error bounds on $\|\widehat{\boldsymbol{\Sigma}} - \boldsymbol{\Sigma}^\star\|_F$ exhibit a natural scale-equivariant property (see Section 4, Scale Equivariance).

## 2.2 The perceptual adjustment query (PAQ)

We assume that every point in our feature space $\mathbb{R}^d$ corresponds to some item. Recall from Figure 1 that a PAQ collects similarity data between a pair of items, where a reference item is fixed, and a spectrum of target items is generated from a one-dimensional path in the feature space. Denote the reference item by $\boldsymbol{x} \in \mathbb{R}^d$. The target items can be generated by any path in $\mathbb{R}^d$, but, for simplicity, we consider straight lines. For any vector $\boldsymbol{a} \in \mathbb{R}^d$, we construct the line $\{\boldsymbol{x} + \gamma\boldsymbol{a} : \gamma \in [0, \infty)\}$. We call this vector $\boldsymbol{a}$ the *query vector*. As shown in Figure 3, the user moves the slider from left to right, and the value of $\gamma$ increases proportionally to the distance traversed by the slider. Note that the value $\gamma$ is *dimensionless*.

The user is instructed to stop the slider at the transition point where the target item transitions from being similar to dissimilar with the reference item. According to our model, this transition point occurs when the $\boldsymbol{\Sigma}^\star$-Mahalanobis distance between the target item and the reference item is $y$. The (noiseless) transition point, denoted by $\gamma_\star$, satisfies the equation

$$y = \|\boldsymbol{x} - (\boldsymbol{x} + \gamma_\star\boldsymbol{a})\|_{\boldsymbol{\Sigma}^\star}^2 = \gamma_\star^2 \boldsymbol{a}^\top \boldsymbol{\Sigma}^\star \boldsymbol{a}. \tag{1}$$

Note that the ideal PAQ response $\gamma_\star$ does not depend on the specific reference item $\boldsymbol{x}$ but rather only on the query direction $\boldsymbol{a}$ and the (unknown) metric matrix $\boldsymbol{\Sigma}^\star$. When querying users with PAQs, the practitioner has control over how the query vectors $\boldsymbol{a}$ are selected, which we discuss in Section 3.2.

## 2.3 Noise model

We model the noise in human responses as follows: In the PAQ response (1), we replace the boundary value $y$ by $y + \eta$, where $\eta \in \mathbb{R}$ represents noise. Thus the user provides a noisy response $\gamma$ whose value satisfies $\gamma^2 \boldsymbol{a}^\top \boldsymbol{\Sigma}^\star \boldsymbol{a} = y + \eta$. Substituting in (1), we have $\gamma^2 = \gamma_\star^2 + \frac{\eta}{\boldsymbol{a}^\top \boldsymbol{\Sigma}^\star \boldsymbol{a}}$. This model captures qualitatively how we would expect the variance of $\gamma$ due to noise to scale. To see why, recall that $\gamma$ is proportional to the distance traversed by the slider in the user interface Figure 1 (bottom). If $\boldsymbol{a}^\top \boldsymbol{\Sigma}^\star \boldsymbol{a}$ is large, then the semantic meaning of the item changes rapidly as the user moves the slider, and the slider will stop at a position that is close to the true transition point. On the other hand, if $\boldsymbol{a}^\top \boldsymbol{\Sigma}^\star \boldsymbol{a}$ is small, then the item changes slowly as the user moves the slider. It is then hard to determine where exactly the transition occurs, so the slider may stop in a larger interval around the transition point.

# 3 Methodology

In this section, we formally present the statistical estimation problem for metric learning from noisy PAQ data, and we develop our algorithm for estimating the true metric matrix $\boldsymbol{\Sigma}^\star$.

## 3.1 Statistical estimation

Assume we collect $N$ PAQ responses, using $N$ query vectors $\{\boldsymbol{a}_i\}_{i=1}^N$ that we select[3]. Denote the noise associated with these queries by random variables $\eta_1, \ldots, \eta_N \in \mathbb{R}$. We obtain PAQ responses, denoted by $\gamma_1, \ldots, \gamma_N$, that satisfy

$$\gamma_i^2 \boldsymbol{a}_i^\top \boldsymbol{\Sigma}^\star \boldsymbol{a}_i = y + \eta_i, \quad i = 1, \ldots, N. \tag{2}$$

We assume the noise variable $\eta$ is independent[4] of the query $\boldsymbol{a}$, has zero mean and variance $\nu_\eta^2$, and is bounded, with $-y \le \eta \le \eta^\uparrow$ for some constant $\eta^\uparrow \ge 0$. Note that we must have $\eta + y \ge 0$ since $\gamma^2 \ge 0$; in addition, we place an upper bound $\eta^\uparrow$ on the noise.

---

[3] In what follows, we use the terms "responses"/"measurements" interchangeably for $\gamma$, and the terms "query vector"/"sensing vector" interchangeably for $\boldsymbol{a}$.

[4] This could be relaxed by placing conditions on the *conditional* distributions of $\eta$ given $\boldsymbol{a}$ (and even the reference point $\boldsymbol{x}$), but we omit this for simplicity.

Given the query directions $\{a_i\}_{i=1}^N$ and the PAQ responses $\{\gamma_i\}_{i=1}^N$, we want to estimate the matrix $\boldsymbol{\Sigma}^\star$. We first rewrite our measurement model as follows: Recall that the matrix inner product is denoted by $\langle \boldsymbol{A}, \boldsymbol{B} \rangle := \mathrm{tr}\left(\boldsymbol{A}^\top \boldsymbol{B}\right)$ for any two matrices $\boldsymbol{A}$ and $\boldsymbol{B}$ of compatible dimension. Then from (2), we write

$$\gamma^2 = \frac{y + \eta}{\boldsymbol{a}^\top \boldsymbol{\Sigma}^\star \boldsymbol{a}}. \tag{3}$$

Plugging (3) once more into (2), we have

$$y + \eta = \langle \boldsymbol{A}^{\mathrm{inv}}, \boldsymbol{\Sigma}^\star \rangle,$$

where

$$\boldsymbol{A}^{\mathrm{inv}} := \gamma^2 \boldsymbol{a}\boldsymbol{a}^\top = \frac{y + \eta}{\boldsymbol{a}^\top \boldsymbol{\Sigma}^\star \boldsymbol{a}} \boldsymbol{a}\boldsymbol{a}^\top. \tag{4}$$

Hence, our problem resembles trace regression, and, in particular, low-rank matrix estimation from rank-one measurements (because the matrix $\boldsymbol{A}^{\mathrm{inv}}$ has rank 1) [31, 30, 32, 33]. We call $\boldsymbol{A}^{\mathrm{inv}}$ the sensing matrix, and $\boldsymbol{a}$ the sensing vector. Classical trace regression assumes that we make (noisy) observations of the form $y = \langle \boldsymbol{A}, \boldsymbol{\Sigma}^\star \rangle + \epsilon$ where $\boldsymbol{A}$ is fixed before we make the measurement; in our problem, the sensing matrix $\boldsymbol{A}^{\mathrm{inv}}$ depends on our observed response $\gamma$ and associated sensing vector $\boldsymbol{a}$. Hence, the process of obtaining a PAQ response can be viewed as an *inversion* of the standard trace measurement process. The inverse nature of our problem makes estimator design more challenging, as we discuss in the following section.

### 3.2 Algorithm

As our first attempt at a procedure to estimate $\boldsymbol{\Sigma}^\star$, we follow the literature [24, 33] and consider randomly sampling i.i.d. vectors $\boldsymbol{a}_i \sim \mathcal{N}(\boldsymbol{0}, \boldsymbol{I}_d)$. We then use standard least-squares estimation of $\boldsymbol{\Sigma}^\star$. Since we expect $\boldsymbol{\Sigma}^\star$ to be low-rank, we add nuclear-norm regularization to promote low rank. In particular, we solve the following program:

$$\min_{\boldsymbol{\Sigma} \succeq \boldsymbol{0}} \frac{1}{N} \sum_{i=1}^N \left(y - \langle \boldsymbol{A}_i^{\mathrm{inv}}, \boldsymbol{\Sigma} \rangle\right)^2 + \lambda_N \|\boldsymbol{\Sigma}\|_*, \tag{5}$$

where $\lambda_N > 0$ is a regularization parameter. This is a convex semidefinite program and can be solved with standard off-the-shelf solvers. However, the inverted form of our measurement model creates two critical issues when naïvely using (5):

- **Bias of standard matrix estimators due to dependence.** Note that the sensing matrix (4) depends on the noise $\eta$. Quantitatively, we have $\mathbb{E}\left[\eta \boldsymbol{A}^{\mathrm{inv}}\right] \neq \boldsymbol{0}$ (see Appendix D.1). Standard trace regression analyses require that this quantity be zero, typically assuming (at least) that $\eta$ is zero-mean conditioned on the sensing matrix $\boldsymbol{A}$. The failure of this to hold in our case introduces a bias that does not decrease with the sample size $N$.

- **Heavy-tailed sensing matrix.** The factor $\frac{1}{\boldsymbol{a}^\top \boldsymbol{\Sigma}^\star \boldsymbol{a}}$ in $\boldsymbol{A}^{\mathrm{inv}}$ (see Equation (4)) makes $\boldsymbol{A}^{\mathrm{inv}}$ heavy-tailed in general. When $\boldsymbol{a}$ is Gaussian, the term $\frac{1}{\boldsymbol{a}^\top \boldsymbol{\Sigma}^\star \boldsymbol{a}}$ is an inverse weighted chi-square random variable, whose higher-order moments are infinite (and the number of finite moments depends on the rank of $\boldsymbol{\Sigma}^\star$). This makes error analysis more difficult, as standard analyses require the sensing matrix $\boldsymbol{A}$ to concentrate well (e.g., be sub-exponential).

To overcome these challenges, we make two key modifications to the procedure (5).

**Step 1: Bias reduction via averaging.** First, we want to mitigate the bias due to the dependence between the sensing matrix $\boldsymbol{A}^{\mathrm{inv}}$ and the noise $\eta$. The bias term $\mathbb{E}\left[\eta \boldsymbol{A}^{\mathrm{inv}}\right]$ scales proportionally to $\mathbb{E}\left[\eta(y + \eta)\right] = \mathbb{E}\left[\eta^2\right]$. Therefore, to reduce this bias in the least-squares estimator (5), we need to reduce the noise variance. We reduce the effective noise variance (and hence the bias) by *averaging* i.i.d. samples. Operationally, instead of obtaining $N$ measurements from $N$ distinct sensing vectors $\{a_i\}_{i=1}^N$, we draw $n$ sensing vectors $\{a_i\}_{i=1}^n$, and collect $m$ measurements, denoted by $\{\gamma_i^{(j)}\}_{j=1}^m$, corresponding to each sensing vector $a_i$. We refer to $n$ as the number of (distinct) sensing vectors.

---

**Algorithm 1** Inverted measurement sensing, averaging, and truncation.

---

**Input:** number of total measurements $N$, averaging parameter $m$ (that divides $N$), truncation threshold $\tau$, measurement value $y$

 1: Compute the number of sensing vectors $n = \frac{N}{m}$
 2: **for** $i = 1$ **to** $n$ **do**
 3:    Draw sensing vector $\boldsymbol{a}_i$ from standard multivariate normal distribution
 4:    Obtain $m$ PAQ measurements $(\gamma_i^{(1)})^2, \ldots, (\gamma_i^{(m)})^2$ with $\boldsymbol{a}_i$ and $y$.
 5:    Bias elimination via averaging: compute averaged response $\bar{\gamma}_i^2 = \frac{1}{m} \sum_{j=1}^m (\gamma_i^{(j)})^2$.
 6:    Heavy tail mitigation via truncation: compute truncated response $\widetilde{\gamma}_i^2 = \bar{\gamma}_i^2 \wedge \tau$.
 7: **end for**
**Output:** truncated responses $\widetilde{\gamma}_1^2, \ldots \widetilde{\gamma}_n^2$

---

To keep the total number of measurements constant, we set $n = \frac{N}{m}$, where the value of $m$ is specified later. For each sensing vector $\boldsymbol{a}_i$, we compute the empirical mean of the $m$ measurements:

$$\bar{\gamma}_i^2 := \frac{1}{m} \sum_{j=1}^m (\gamma_i^{(j)})^2 = \frac{1}{m} \sum_{j=1}^m \frac{y + \eta_i^{(j)}}{\boldsymbol{a}_i^\top \boldsymbol{\Sigma}^\star \boldsymbol{a}_i} = \frac{y + \bar{\eta}_i}{\boldsymbol{a}_i^\top \boldsymbol{\Sigma}^\star \boldsymbol{a}_i}, \tag{6}$$

where we define the average noise by $\bar{\eta}_i := \frac{1}{m} \sum_{j=1}^m \eta_i^{(j)}$. This averaging operation reduces the effective noise variance from $\mathrm{var}(\eta_i) = \nu_\eta^2$ to $\mathrm{var}(\bar{\eta}_i) = \frac{\nu_\eta^2}{m}$. If $n$ is small, we may have large error due to an insufficient number of query vectors $\boldsymbol{a}_i$. On the other hand, a small $m$ leads to a large bias. Therefore, we set the value of $m$ carefully to balance these two effects. This is studied theoretically in Section 4 and demonstrated empirically in Section 5.

**Step 2: Heavy tail mitigation via truncation.** Next, we need to control the heavy-tailed behavior introduced by the $\frac{1}{\boldsymbol{a}^\top \boldsymbol{\Sigma}^\star \boldsymbol{a}}$ term in the sensing matrix $\boldsymbol{A}^{\mathrm{inv}}$. Note that the sample averaging procedure (6) does not mitigate this problem. We adopt the approach in [39] and truncate the observations. Specifically, we truncate the averaged measurements $\bar{\gamma}_i^2$ to $\widetilde{\gamma}^2 := \bar{\gamma}^2 \wedge \tau$, where $\tau > 0$ is a truncation threshold that we specify later. We then construct the truncated sensing matrices

$$\widetilde{\boldsymbol{A}}_i = \widetilde{\gamma}_i^2 \boldsymbol{a}_i \boldsymbol{a}_i^\top = \left( \frac{y + \bar{\eta}_i}{\boldsymbol{a}_i^\top \boldsymbol{\Sigma}^\star \boldsymbol{a}_i} \wedge \tau \right) \boldsymbol{a}_i \boldsymbol{a}_i^\top, \quad i = 1, \ldots, n. \tag{7}$$

While truncation mitigates heavy-tailed behavior, it also introduces additional bias in our estimate. The truncation threshold $\tau$ therefore gives us another tradeoff, and in our analysis to follow, we carefully set the value of $\tau$ to balance the effects of heavy-tailedness and bias.

**Final algorithm.** Before presenting our final optimization program, we summarize our assumptions and sensing model below.

**Assumption 1.** *The noise values $\eta_i$ are i.i.d copies of the random variable $\eta$, which is independent of the random sensing vector $\boldsymbol{a}$. The random noise is (1) zero-mean: $\mathbb{E}[\eta] = 0$, and (2) bounded: There exists a positive constant $\eta^\uparrow$ such that $-y \leq \eta \leq \eta^\uparrow$ with probability 1.*

We choose the sensing vector distribution to be the standard multivariate normal distribution and collect, average, and truncate $N$ PAQ responses following Algorithm 1. This process yields $n$ truncated responses $\widetilde{\gamma}_1^2, \ldots \widetilde{\gamma}_n^2$. We then use these truncated responses to form the averaged and truncated matrices $\{\widetilde{\boldsymbol{A}}_i\}_{i=1}^n$, which we substitute into the original least-squares problem (5). To estimate $\boldsymbol{\Sigma}^\star$, we solve

$$\widehat{\boldsymbol{\Sigma}} \in \underset{\boldsymbol{\Sigma} \succeq \boldsymbol{0}}{\arg \min} \ \frac{1}{n} \sum_{i=1}^n \left( y - \langle \widetilde{\boldsymbol{A}}_i, \boldsymbol{\Sigma} \rangle \right)^2 + \lambda_n \|\boldsymbol{\Sigma}\|_*, \tag{8}$$

where, again, $\lambda_n$ is a regularization parameter that we specify later.

**Practical considerations.** In the averaging step, we collect $m$ measurements for each sensing vector $\boldsymbol{a}_i$. These measurements could be collected from $m$ different users. Furthermore, recall from Section 2.2 that the measurements do not depend on the reference item $\boldsymbol{x}$. As a result, one may

also collect multiple responses from the same user by presenting the same query vector $\boldsymbol{a}_i$ with different reference items. In addition, recall from Section 2.1 that user responses are scale-invariant. Practitioners are hence free to set the boundary $y$ to be any positive value of their choice without loss of generality, and the noise $\eta$ scales accordingly with $y$. The user interface does not depend on the value of $y$.

## 4  Theoretical results

We now present our main theoretical result, which is a finite-sample error bound for estimating a low-rank metric from inverted measurements with the nuclear norm regularized estimator (8). Our error bound is generally stated, and depends on the averaging parameter $m$ and the truncation threshold $\tau$. Recall that $\nu_\eta^2$ denotes the variance of $\eta$. We define the quantities $y^\uparrow := y + \eta^\uparrow$ and $\mu_y = y + \texttt{median}(\eta)$. We further denote by $\sigma_1 \geq \cdots \geq \sigma_r > 0$ the non-zero singular values of $\boldsymbol{\Sigma}^\star$.

**Theorem 1.** *Suppose $\boldsymbol{\Sigma}^\star$ is rank $r$, with $r > 8$. Assume that we choose the sensing vector distribution the be the standard multivariate normal distribution, that Assumption 1 holds on the noise, and that we collect, average, and truncate measurements following Algorithm 1. Further, assume that the truncation threshold $\tau$ satisfies $\tau \geq \frac{\mu_y}{\mathrm{tr}(\boldsymbol{\Sigma}^\star)}$. Then there are positive constants $c, C, C_1,$ and $C_2$, such that if the regularization parameter and the number of sensing vectors satisfy*

$$\lambda_n \geq C_1 \left[ y^\uparrow \left( \frac{y^\uparrow}{\sigma_r r} \sqrt{\frac{d}{n}} + \frac{d}{n} \tau + \left( \frac{y^\uparrow}{\sigma_r r} \right)^2 \frac{1}{\tau} \right) + \frac{1}{\sigma_r r} \frac{\nu_\eta^2}{m} \right] \quad \text{and} \quad n \geq C_2 rd, \qquad (9)$$

*then any solution $\widehat{\boldsymbol{\Sigma}}$ to the optimization program (8) satisfies*

$$\|\widehat{\boldsymbol{\Sigma}} - \boldsymbol{\Sigma}^\star\|_F \leq C \left( \frac{\mathrm{tr}(\boldsymbol{\Sigma}^\star)}{\mu_y} \right)^2 \sqrt{r} \lambda_n \qquad (10)$$

*with probability at least $1 - 4e^{-d} - e^{-cn}$.*

The proof of Theorem 1 is presented in Appendix E. The two sources of bias discussed in Section 3.2 appear in the expression (9) for the regularization parameter $\lambda_n$ (and consequently in the error bound (10)). The term scaling as $1/\tau$ corresponds to the bias induced by truncation, and decreases as the truncation gets milder (i.e., as the threshold $\tau$ gets larger). The term scaling as $\nu_\eta^2/m$ corresponds to the bias arising from dependence between the noise and sensing matrix. As discussed in Section 3.2, in this model, $m$-averaging results in a bias that scales like $1/m$. Given the dependence of the estimation error bound on the parameters $m$ and $\tau$, we carefully set these parameters to obtain a tight bound as a function of the number of *total measurements* $N = mn$. These choices for $m$ and $\tau$, along with the final estimation error, are presented in the following corollary, proved in Appendix F.

**Corollary 1.** *Recall that $N = mn$. Assume that the conditions of Theorem 1 hold, and set the values of the constants $(c, C, C_1, C_2)$ according to Theorem 1. Suppose that the number of total measurements satisfies*

$$N \geq \left\{ 2C_2^{3/2} \frac{\nu_\eta^2}{(y^\uparrow)^2} r^{3/2} d \right\} \vee \left\{ C_2 rd \right\}. \qquad (11)$$

*Set the averaging parameter $m$ and truncation threshold $\tau$ to be*

$$m = \left\lceil \left( \frac{\nu_\eta^2}{(y^\uparrow)^2} \right)^{2/3} \left( \frac{N}{d} \right)^{1/3} \right\rceil \quad \text{and} \quad \tau = \frac{y^\uparrow}{\sigma_r r} \sqrt{\frac{N}{md}}, \qquad (12)$$

*and set $\lambda_n$ equal to its lower bound in (9). With probability at least $1 - 4e^{-d} - e^{-cN/m}$, we have:*

*(a) If $\frac{\nu_\eta^2}{(y^\uparrow)^2} > \sqrt{\frac{d}{N}}$, then any solution $\widehat{\boldsymbol{\Sigma}}$ to the optimization program (8) satisfies*

$$\|\widehat{\boldsymbol{\Sigma}} - \boldsymbol{\Sigma}^\star\|_F \leq C' \frac{\sigma_1^2}{\sigma_r} \frac{(y^\uparrow)^{4/3}(\nu_\eta^2)^{1/3}}{\mu_y^2} r^{3/2} \left( \frac{d}{N} \right)^{1/3}. \qquad (13)$$

*(b) If $\frac{\nu_\eta^2}{(y^\uparrow)^2} \le \sqrt{\frac{d}{N}}$, then any solution $\widehat{\Sigma}$ to the optimization program* (8) *satisfies*

$$\|\widehat{\Sigma} - \Sigma^\star\|_F \le C' \frac{\sigma_1^2}{\sigma_r} \left(\frac{y^\uparrow}{\mu_y}\right)^2 r^{3/2} \left(\frac{d}{N}\right)^{1/2}. \tag{14}$$

*In both cases, $C' = 3C \cdot C_1$.*

A few comments are warranted about our error bounds (13) and (14):

**Error rates and noise regimes.** Under the standard trace measurement model, it is known that if the measurement matrices are i.i.d. according to some sub-Gaussian distribution and the number of measurements satisfies $N \gtrsim rd$, then nuclear norm regularized estimators achieve an error that scales like $\sqrt{\frac{rd}{N}}$ (e.g., [24, 25]). Such a result is also known to be minimax optimal [25]. Allowing heavier-tailed assumptions on the sensing matrices, such as sub-exponential [32, 42] or bounded fourth moment [39], typically results in additional $\log d$ factors but does not impact the exponent $1/2$ in the error rate. However, a crucial assumption in these results is that $\mathbb{E}\left[\eta A^{\text{inv}}\right] = 0$, and thus there is no bias due to measurement noise. Our inverted measurement sensing matrix is not only heavy-tailed but also leads to bias (see Lemma 1 in Appendix D.1). Nevertheless, we are able to reduce the bias and trade it for variance, ensuring consistent estimation in all regimes.

In Corollary 1, there are two distinct cases for error rate which correspond to two different noise regimes induced by the quantity $\nu_\eta^2/(y^\uparrow)^2$, which captures the noise level in our measurements. In particular, the two cases in Corollary 1 correspond to two regimes with distinct bias behavior:

  (a) High-noise regime: In this setting, the bias due to measurement noise is non-negligible. As a result, we employ averaging with large $m$, which results in the rate scaling as $(d/N)^{1/3}$.

  (b) Low-noise regime: In this setting, the measurement noise bias is dominated by the variance, and thus has negligible impact on the estimation error. As a result, we are able to achieve a rate of order $(d/N)^{1/2}$, which is consistent with established results for low-rank matrix estimation.

**Sample complexity.** Since the degrees of freedom in a rank-$r$ matrix of size $d \times d$ is of order $rd$, one expects that the minimum number of measurements to identify a rank-$r$ matrix is of order $rd$. This is reflected in Theorem 1, which assumes that the number of *distinct* sensing vectors $\{a_i\}$ satisfies $n \gtrsim rd$. In the high-noise regime, from (12) in Corollary 1, we have that $m$ scales like $(N/d)^{1/3}$. Thus, the total number of measurements is $N = mn \gtrsim (N/d)^{1/3} \cdot rd \gtrsim N^{1/3} d^{2/3} r$, and hence $N \gtrsim r^{3/2} d$. Given that the rank is assumed to be relatively small compared to the dimension, the extra factor of $\sqrt{r}$ is a relatively small price to pay to obtain consistent estimation. In the low-noise regime, it can be verified that $m = 1$ in (12) due to the low-noise condition $\nu_\eta^2/(y^\uparrow)^2 \le \sqrt{d/N}$. No averaging is needed, and we only require $N = n \gtrsim rd$.

**Dependence on rank.** When compared to standard results, Corollary 1 differs in its dependence on rank. First, the matrix $\Sigma^\star$ is assumed to have rank $r > 8$. This prevents the term $\frac{1}{a^\top \Sigma^\star a}$ from making the sensing matrices so heavy-tailed that even truncation does not help. We empirically show that the assumption of $r > 8$ is necessary in Section 5. Second, there is an additional factor of $r$ in our rate for both noise regimes. To interpret this, note that if $\Sigma^\star$ has non-zero singular values in a fixed range, then $\mathbb{E}\left[a^\top \Sigma^\star a\right] = \text{tr}\left(\Sigma^\star\right) \approx r$. Since the "magnitude" of the sensing matrix $A^{\text{inv}}$ is inversely proportional to $a^\top \Sigma^\star a$, increasing $r$ decreases the magnitude of $A^{\text{inv}}$ and thus also (for a fixed noise level) the signal-to-noise ratio.

**Scale equivariance.** As discussed in Section 2.1, the metric learning from PAQs problem aims to find an equivalence class $\{(c_{\text{scale}}\Sigma, c_{\text{scale}}y) : c_{\text{scale}} > 0\}$, and the ground-truth $\Sigma^\star$ is defined with respect to a particular choice of $y$. Accordingly, our error bounds are scale-equivariant: If we instead replaced $y$ with $c_{\text{scale}}y$, the bounds (13) and (14) would scale linearly in $c_{\text{scale}}$. This fact is precisely verified in Appendix C and relies on the fact that the noise also scales appropriately in $c_{\text{scale}}$. Thus practitioners may simply set $y$ to be *any* positive number.

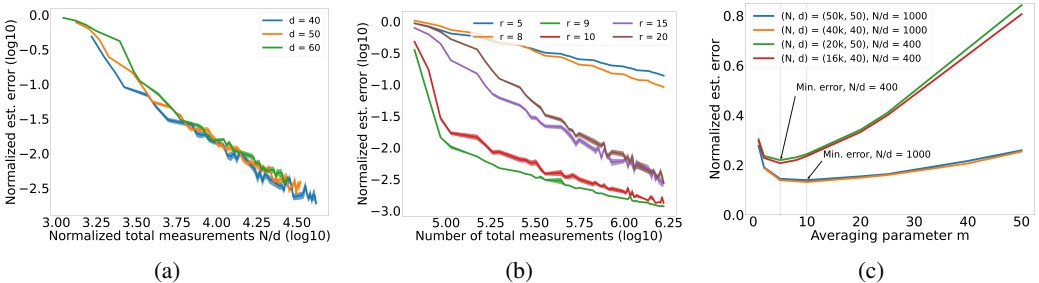

Figure 4: Simulations quantifying the effect of dimension $d$, rank $r$, and averaging parameter $m$ on estimation error. Shaded areas correspond to standard error of the mean but sometimes not visible.

## 5 Numerical simulations

In this section, we provide numerical simulations investigating the effects of the various problem and estimation parameters. For all results, we report the normalized estimation error $\|\widehat{\Sigma} - \Sigma^{\star}\|_F / \|\Sigma^{\star}\|_F$ averaged over 20 trials. Shaded areas (sometimes not visible) represent standard error of the mean. For all experiments, we follow [19] and generate the ground-truth metric matrix as $\Sigma^{\star} = \frac{d}{\sqrt{r}} U U^{\top}$, where $U \in \mathbb{R}^{d \times r}$ is a randomly generated matrix with orthonormal columns. The noise $\eta$ is sampled from a uniform distribution on $[-\eta^{\uparrow}, \eta^{\uparrow}]$ (where $\eta^{\uparrow} \leq y$). We set the regularization parameter, truncation threshold, and averaging parameter in a manner consistent with our theoretical results (see Eqs. (9) and (12)), cross-validating to choose the constant factors. We solve the optimization problem using `cvxpy` [43, 44]. Code for all simulations is provided at `https://github.com/austinxu87/paq`.

**Effects of dimension and rank.** Our first set of experiments characterizes the effects of dimension $d$ and matrix rank $r$. For all experiments, unless we are sweeping a specific parameter, we set $y = 200$, $d = 50$, $r = 15$, and $\eta^{\uparrow} = 10$. Fig. 4a shows the performance for varying values of $d$ plotted against the normalized sample size $N/d$. For all dimensions $d$, the error decays to zero as the total number of measurements $N$ increases. Furthermore, the error curves are well-aligned when the sample size is normalized by $d$ with fixed $r$, empirically aligning with Corollary 1. Fig. 4b shows the performance for varying values of rank $r$. Recall that for our theoretical results we assume $r > 8$ to ensure that the quadratic term $a^{\top} \Sigma^{\star} a$ in the denominator of our sensing matrices does not lead to excessively heavy-tailed behavior. When $r > 8$, the number of measurements required for the same estimation error increases as the rank increases. A clear phase transition occurs at $r = 8$. The error still decreases with $N$ for $r \leq 8$, but at a markedly slower rate than when $r > 8$. This empirically demonstrates that when $r \leq 8$, the sensing matrix tails are too heavy to be mitigated by truncation.

**Effect of averaging parameter $m$.** Equation (12) suggests that the averaging parameter $m$ should scale proportionally to $(N/d)^{1/3}$. To test this, we set $y = 200$, $d = 50$, $r = 9$, and $\eta^{\uparrow} = 200$. We vary values of $m$ for different choices of the $(N, d)$ pair, as shown in Fig. 4c. The empirically optimal choice of $m$ is observed to be the same when $N/d$ is fixed, regardless of the particular choices of $N$ or $d$ (the green and red curves overlap, and the blue and orange curves overlap). Moreover, the optimal $m$ is smaller when $N/d = 400$ compared to when $N/d = 1000$.

## 6 Discussion

In this paper we introduced the perceptual adjustment query, a cognitively lightweight way to obtain expressive responses from humans. We specifically investigate using PAQs for human perceptual similarity learning. We use a Mahalanobis distance-based model for human similarity perception and use PAQs to estimate the unknown metric. Using random measurements to learn an unknown Mahalanobis metric gives rise to the new inverted measurement scheme for high dimensional low rank matrix estimation which violates commonly held assumptions for existing estimators. We developed a two-stage estimator and provide corresponding sample complexity guarantees. This work opens up many interesting lines of future work in inverted measurements. One key direction is to characterize their fundamental limits with information-theoretic lower bounds.

## Acknowledgments

AX was supported by National Science Foundation grant CCF-2107455. JW was supported by the Ronald J. and Carol T. Beerman President's Postdoctoral Fellowship and the Algorithms and Randomness Center Postdoctoral Fellowship at Georgia Tech. MD was supported by National Science Foundation grants CCF-2107455, IIS-2212182, and DMS-2134037. AP was supported in part by the National Science Foundation grants CCF-2107455 and DMS-2210734, and by research awards/gifts from Adobe, Amazon and Mathworks.

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

## A  Broader impacts and limitations

In this section, we discuss the broader impacts and limitations of our work.

### A.1  Broader impacts

Automated systems that use human feedback are being used in an increasing number of contexts, spanning everything from predicting user preferences to finetuning language models. It is important to ensure that such systems are as accurate as possible; this naturally requires humans to respond in an accurate and consistent manner. Using the perceptual adjustment query to collect data in such settings could lead to more expressive responses without heavy cognitive burdens on users. Furthermore, providing a user the additional context of a continuous spectrum of items may result in more self-consistent responses. The downstream effects of collecting more expressive and self-consistent human responses could lead to improved models or entirely new paradigms of model development for a myriad of problem settings. With these advantages come associated risks as well. Due to how expressive the responses to PAQs are, the effects of adversarial responses may be magnified. That is, if an adversary purposely chooses to respond in an antagonistic manner, models trained with PAQs may be trained poorly or in opposition to the stated goal. Mitigating such effects likely requires a holistic approach from both the query design and robust model design perspectives.

### A.2  Limitations

From a data collection perspective, the perceptual adjustment query requires access to a continuous space where each point corresponds to an item. In many applications, assuming access to this continuous space is reasonable. For example, if we use PAQs to characterize color blindness, then a natural continuous space is the RGB color space. In general, we situate our data collection within the latent space of a generative model, such as a GAN. While GANs are capable of producing extremely high fidelity images, these images are not always free of semantically meaningful artifacts. Our query design and modeling assumptions do not explicitly consider the case where a portion of the continuous spectrum of items may be corrupted. Furthermore, because our work is an initial exploration into low-rank matrix estimation from inverted measurements, we have not considered scenarios such as unbounded noise or heavier-tailed sensing vectors, and we have not established information-theoretic lower bounds for the inverted measurement paradigm. We hope that further exploration of the inverted measurement paradigm will lead to a rich line of follow-up work.

## B  Simulation details

In this section, we provide details for the simulation results presented in Figure 2. For our experiments, we adopt a normalized version of the setup of [30] and form the metric $\boldsymbol{\Sigma}^\star$ by $\boldsymbol{\Sigma}^\star = \boldsymbol{L}\boldsymbol{L}^\top / \|\boldsymbol{L}\boldsymbol{L}^\top\|_F$, where $\boldsymbol{L}$ is a $50 \times 10$ matrix with i.i.d. Gaussian entries. We sweep the number of query responses $N$, estimate the metric with $\widehat{\boldsymbol{\Sigma}}$, and report the normalized estimation error $\|\boldsymbol{\Sigma}^\star - \widehat{\boldsymbol{\Sigma}}\|_F / \|\boldsymbol{\Sigma}^\star\|_F$ averaged over 10 independent trials. For each query response, items are drawn i.i.d. from a standard multivariate normal distribution, similar to [45].

**Pairwise comparison setup.**  For pairwise comparisons, we use value of $y = 10$ to denote the squared distance at which items become dissimilar, following our distance-based model for human perception (see Section 2.1). For the $i$-th pairwise comparisons, we draw two items $\boldsymbol{x}_1^{(i)}, \boldsymbol{x}_2^{(i)}$ i.i.d. from a standard multivariate normal distribution. We record the pairwise comparison outcome $\epsilon_i \in \{-1, +1\}$ as $\epsilon_i = \text{sign}(\|\boldsymbol{x}_1^{(i)} - \boldsymbol{x}_2^{(i)}\|_{\boldsymbol{\Sigma}^\star}^2 - y)$. To estimate the metric from pairwise comparisons, we utilize a nuclear-norm regularized hinge loss and solve the following optimization problem:

$$\widehat{\boldsymbol{\Sigma}}_{\text{PC}} \in \underset{\boldsymbol{\Sigma} \succeq \boldsymbol{0}}{\arg\min} \ \frac{1}{N} \sum_{i=1}^{N} \max\{0, y - \epsilon_i \|\boldsymbol{x}_1^{(i)} - \boldsymbol{x}_2^{(i)}\|_{\boldsymbol{\Sigma}}^2\} + \lambda_{\text{PC}} \|\boldsymbol{\Sigma}\|_*.$$

**Triplet setup.**  For the $i$-th triplet, we draw three items $\boldsymbol{x}_1^{(i)}, \boldsymbol{x}_2^{(i)}, \boldsymbol{x}_3^{(i)}$ i.i.d. from a standard multivariate normal distribution and record the outcome $\epsilon_i \in \{-1, +1\}$ as $\epsilon_i = \text{sign}(\|\boldsymbol{x}_1^{(i)} -$

$\boldsymbol{x}_2^{(i)}\|_{\boldsymbol{\Sigma}^\star}^2 - \|\boldsymbol{x}_1^{(i)} - \boldsymbol{x}_3^{(i)}\|_{\boldsymbol{\Sigma}^\star}^2)$. To estimate the metric from triplet responses, we follow [19] and utilize a nuclear-norm regularized hinge loss and solve the following optimization problem:

$$\widehat{\boldsymbol{\Sigma}}_\mathrm{T} \in \operatorname*{arg\,min}_{\boldsymbol{\Sigma} \succeq \mathbf{0}} \frac{1}{N} \sum_{i=1}^N \max\left\{0, 1 - \epsilon_i\left(\|\boldsymbol{x}_1^{(i)} - \boldsymbol{x}_2^{(i)}\|_{\boldsymbol{\Sigma}}^2 - \|\boldsymbol{x}_1^{(i)} - \boldsymbol{x}_3^{(i)}\|_{\boldsymbol{\Sigma}}^2\right)\right\} + \lambda_\mathrm{T}\|\boldsymbol{\Sigma}\|_*.$$

**Ranking-$k$ query setup.** For the $i$-th ranking query with a reference item $\boldsymbol{x}_0$ and $k$ items $\boldsymbol{x}_1, \ldots, \boldsymbol{x}_k$ to be ranked, we draw all items i.i.d. from a standard multivariate normal distribution. For each item $\boldsymbol{x}_k$, we compute the squared distance $\|\boldsymbol{x}_0 - \boldsymbol{x}_k\|_{\boldsymbol{\Sigma}^\star}^2$. To determine the ranking of items, we sort the items based on this squared distance. To estimate the metric, we follow the approach of [14] and decompose the full ranking into its constituent triplets. A ranking consisting of $k$ items can equivalently be decomposed into $k(k-1)/2$ triplet responses. To estimate the metric, we decompose each ranking query and use the triplet estimator presented above with regularization parameter $\lambda_\mathrm{R}$ to obtain estimate $\widehat{\boldsymbol{\Sigma}}_{\mathrm{R}\text{-}k}$.

**PAQ setup.** For the $i$-th PAQ response, we draw the reference item $\boldsymbol{x}_i$ and query vector $\boldsymbol{a}_i$ i.i.d. from the standard multivariate normal distribution. We then receive a scaling $\gamma_i^2$ satisfying $\gamma_i^2 = y/\boldsymbol{a}_i^\top \boldsymbol{\Sigma}^\star \boldsymbol{a}_i$, with $y = 10$. To perform estimation, we leverage our method presented in Section 3. Our theoretical results indicate that the averaging parameter $m$ should be set to 1 in the noiseless setting. Furthermore, the truncation threshold $\tau$ is large relative to our responses $\gamma_i^2$, meaning no truncation is employed. As a result, we solve the nuclear-norm regularized trace regression problem

$$\widehat{\boldsymbol{\Sigma}}_\mathrm{PAQ} \in \operatorname*{arg\,min}_{\boldsymbol{\Sigma} \succeq \mathbf{0}} \frac{1}{N} \sum_{i=1}^N \left(\langle \boldsymbol{a}_i \boldsymbol{a}_i^\top, \boldsymbol{\Sigma}\rangle - \frac{y}{\gamma_i^2}\right)^2 + \lambda_\mathrm{PAQ}\|\boldsymbol{\Sigma}\|_*.$$

In all cases above, we solve all optimization problems with `cvxpy` and normalize the estimated metric $\widehat{\boldsymbol{\Sigma}}_{\{\mathrm{PC,\,T,\,R}\text{-}k,\,\mathrm{PAQ}\}}$ to be unit Frobenius norm to ensure consistent scaling when compared against the true metric $\boldsymbol{\Sigma}^\star$. We use a value of $0.05$ for all regularization parameters $\lambda_{\{\mathrm{PC,\,T,\,R}\text{-}k,\,\mathrm{PAQ}\}}$ and observe similar performance trends for other choices of regularization parameter.

## C  Scale equivariance

In this section, we verify the scale-equivariance of our derived theoretical bounds (13) and (14). Specifically, we denote by $\boldsymbol{\Sigma}^\star$ and $\widehat{\boldsymbol{\Sigma}}$ the ground-truth and the estimated matrices corresponding to value $y$. We denote by $\boldsymbol{\Sigma}_c^\star$ and $\widehat{\boldsymbol{\Sigma}}_c$ the ground-truth and estimated matrices corresponding to value $c_\mathsf{scale} y$ for any $c_\mathsf{scale} > 0$. By definition, we have $\boldsymbol{\Sigma}_c^\star = c_\mathsf{scale} \boldsymbol{\Sigma}^\star$, and it can be verified that solving the optimization program (8) yields $\widehat{\boldsymbol{\Sigma}}_c = c_\mathsf{scale} \widehat{\boldsymbol{\Sigma}}$. Hence, one expects the error bound to scale as $c_\mathsf{scale}$. To verify this linear scaling in $c_\mathsf{scale}$, we confirm that the noise $\eta$ scales as $c_\mathsf{scale}$.

Under the ground-truth metric $\boldsymbol{\Sigma}^\star$, if the user responds with an item that is a distance $y + \eta$ away from the reference item, then that same item is a distance $c_\mathsf{scale}(y + \eta)$ away from the reference under the scaled setting. As a result, the noise scales as a result of the choice of $y$. Therefore, the following values in the upper bound (13) can be written as scaled versions of their corresponding "ground-truth" values.

| Noise | $\eta = c_\mathsf{scale}\,\eta_\star$ | Noise median | $\mu_y = c_\mathsf{scale}\,\mu_y^\star$ |
|---|---|---|---|
| Noise upper bound | $\eta^\uparrow = c_\mathsf{scale}\,\eta_\star^\uparrow$ | Boundary upper bound | $y^\uparrow = c_\mathsf{scale}\,(y_\star + \eta_\star^\uparrow)$ |
| Noise variance | $\nu_\eta^2 = c_\mathsf{scale}^2\,\nu_{\eta,\star}^2$ | Singular values | $\sigma_k = c_\mathsf{scale}\,\sigma_{k,\star}^\star, k = 1,\ldots,r$ |

Substituting these scaled expressions into the upper bounds (13) and (14), we have

$$\|\widehat{\boldsymbol{\Sigma}}_c - \boldsymbol{\Sigma}_c^\star\|_F \leq c_\mathsf{scale}\, C' \, \frac{(\sigma_1^\star)^2}{\sigma_r^\star} \frac{(y_\star^\uparrow)^{4/3}(\nu_{\eta,\star}^2)^{1/3}}{(\mu_y^\star)^2} \, r^{3/2} \left(\frac{d}{N}\right)^{1/3}$$

in the high-noise regime and

$$\|\widehat{\boldsymbol{\Sigma}}_c - \boldsymbol{\Sigma}_c^\star\|_F \leq c_{\mathsf{scale}}\, C'\, \frac{(\sigma_1^\star)^2}{\sigma_r^\star} \left(\frac{y_\star^\uparrow}{\mu_y^\star}\right)^2 r^{3/2} \left(\frac{d}{N}\right)^{1/2}$$

in the low-noise regime. Note that the constant $C'$ is independent of $c_{\mathsf{scale}}$.

# D  Preliminaries and notation

In this section, we provide an overview of the key tools that are utilized in our proofs. We first introduce notation which is used throughout our proofs.

**Notation.**  For two real numbers $a$ and $b$, let $a \wedge b = \min\{a, b\}$. Given a vector $\boldsymbol{x} \in \mathbb{R}^d$, denote $\|\boldsymbol{x}\|_1$ and $\|\boldsymbol{x}\|_2$ as the $\ell_1$ and $\ell_2$ norm, respectively. Denote $\mathcal{S}^{d-1} := \{x \in \mathbb{R}^d : \|x\|_2 = 1\}$ to be the set of vectors with unit $\ell_2$ norm. Given a matrix $\boldsymbol{A} \in \mathbb{R}^{d_1 \times d_2}$, denote $\|\boldsymbol{A}\|_F$, $\|\boldsymbol{A}\|_*$, and $\|\boldsymbol{A}\|_{\mathrm{op}}$ as the Frobenius norm, nuclear norm, and operator norm, respectively. Denote $\mathbb{S}^{d \times d} = \{\boldsymbol{A} \in \mathbb{R}^{d \times d} : \boldsymbol{A} = \boldsymbol{A}^\top\}$ to be the set of symmetric $d \times d$ matrices. Denote $\boldsymbol{A} \succeq \boldsymbol{0}$ to mean $\boldsymbol{A}$ is symmetric positive semi-definite. For $\boldsymbol{A} \succeq \boldsymbol{0}$, define the (pseudo-) inner product $\langle \boldsymbol{x}, \boldsymbol{y} \rangle_{\boldsymbol{A}} = \boldsymbol{x}^\top \boldsymbol{A} \boldsymbol{y}$ and the associated (pseudo-) norm $\|\boldsymbol{x}\|_{\boldsymbol{A}} = \sqrt{\boldsymbol{x}^\top \boldsymbol{A} \boldsymbol{x}}$. For matrices $\boldsymbol{A}, \boldsymbol{B} \in \mathbb{R}^{d_1 \times d_2}$, denote $\langle \boldsymbol{A}, \boldsymbol{B} \rangle = \mathrm{tr}\left(\boldsymbol{A}^\top \boldsymbol{B}\right)$ as the Frobenius inner product.

We use the notation $f(x) \lesssim g(x)$ to denote that there exists some universal positive constant $c > 0$, such that $f(x) \leq c \cdot g(x)$, and use the notation $f(x) \gtrsim g(x)$ when $g(x) \lesssim f(x)$.

We define random matrices

$$\bar{\boldsymbol{A}} = \bar{\gamma}^2 \boldsymbol{a}\boldsymbol{a}^\top = \frac{y + \bar{\eta}}{\boldsymbol{a}^\top \boldsymbol{\Sigma}^\star \boldsymbol{a}} \boldsymbol{a}\boldsymbol{a}^\top \tag{15}$$

and

$$\widetilde{\boldsymbol{A}} = \widetilde{\gamma}^2 \boldsymbol{a}\boldsymbol{a}^\top = \left(\frac{y + \bar{\eta}}{\boldsymbol{a}^\top \boldsymbol{\Sigma}^\star \boldsymbol{a}} \wedge \tau\right) \boldsymbol{a}\boldsymbol{a}^\top \tag{16}$$

as the sensing matrix formed with the $m$-averaged responses $\bar{\gamma}$ and truncated responses $\widetilde{\gamma}$, respectively.

## D.1  Inverted measurement sensing matrices result in estimation bias

Recall from Equation (4) that the random sensing matrix $\boldsymbol{A}^{\mathsf{inv}}$ takes the form

$$\boldsymbol{A}^{\mathsf{inv}} = \frac{y + \eta}{\boldsymbol{a}^\top \boldsymbol{\Sigma}^\star \boldsymbol{a}} \boldsymbol{a}\boldsymbol{a}^\top.$$

Standard trace regression analysis assumes that for some sensing matrix $\boldsymbol{A}$ and measurement noise $\eta$, $\mathbb{E}[\eta \boldsymbol{A}] = \boldsymbol{0}$. Specifically, it is often typically assumed that $\eta$ is zero-mean conditioned on the sensing matrix $\boldsymbol{A}$. The following lemma shows that for the inverted measurements, we have $\mathbb{E}[\eta \boldsymbol{A}^{\mathsf{inv}}] \neq 0$, resulting in bias in estimation.

**Lemma 1.** *Let $\boldsymbol{A}^{\mathsf{inv}}$ be the random matrix defined in Eq.* (4) *and $\eta$ be the measurement noise. Then*

$$\mathbb{E}\left[\eta \boldsymbol{A}^{\mathsf{inv}}\right] \neq \boldsymbol{0}.$$

The proof of Lemma 1 is provided in Appendix D.6.1. Hence, utilizing established low-rank matrix estimators for inverted measurements results in biased estimation.

## D.2  Sub-exponential random variables

Our analysis utilizes properties of sub-exponential random variables, a class of random variables with heavier tails than the Gaussian distribution.

**Lemma 2** (Moment bounds for sub-exponential random variables [46, Proposition 2.7.1(b)])**.** *If $X$ is a sub-exponential random variable, then there exists some constant $c$ (only dependent on the distribution of the random variable $X$) such that for all integers $p \geq 1$,*

$$\left(\mathbb{E}|X|^p\right)^{1/p} \leq cp.$$

### D.3 Bernstein's inequality

In our proofs, we use Bernstein's inequality to bound the sums of independent sub-exponential random variables.

**Lemma 3** (Bernstein's inequality, adapted from [47, Theorem 2.10])**.** *Let $X_1, \ldots, X_n$ be independent real-valued random variables. Assume there exist positive numbers $u_1$ and $u_2$ such that*

$$\mathbb{E}\left[X_i^2\right] \le u_1 \quad and \quad \mathbb{E}\left[|X_i|^p\right] \le \frac{p!}{2} u_1 u_2^{p-2} \text{ for all integers } p \ge 2,$$

*Then for all $t > 0$,*

$$\mathbb{P}\left(\left|\frac{1}{n}\sum_{i=1}^{n}(X_i - \mathbb{E}\left[X_i\right])\right| \ge \sqrt{\frac{2u_1 t}{n}} + \frac{u_2 t}{n}\right) \le 2\exp(-t).$$

### D.4 Moments of the ratios of quadratic forms

The quadratic term $\boldsymbol{a}^\top \boldsymbol{\Sigma}^\star \boldsymbol{a}$ appears in the denominator of our sensing matrices, so we use the following result to quantify the moments of the ratios of quadratic forms.

**Lemma 4.** *There exists an absolute constant $c > 0$ such that the following is true. Let $\boldsymbol{a} \sim \mathcal{N}(\boldsymbol{0}, \boldsymbol{I}_d)$, $\boldsymbol{\Sigma}^\star \in \mathbb{R}^{d \times d}$ be any PSD matrix with rank $r$, and $\boldsymbol{U} \in \mathbb{R}^{d \times d}$ be an arbitrary symmetric matrix.*

  *(a) Suppose that $r > 8$. Then we have*

$$\mathbb{E}\left(\frac{1}{\boldsymbol{a}^T \boldsymbol{\Sigma}^\star \boldsymbol{a}}\right)^4 \le \frac{c}{\sigma_r^4 r^4}.$$

  *(b) Suppose that $r > 2$. Then we have*

$$\mathbb{E}\left(\frac{\boldsymbol{a}^\top \boldsymbol{U} \boldsymbol{a}}{\boldsymbol{a}^\top \boldsymbol{\Sigma}^\star \boldsymbol{a}}\right) \le \frac{c}{\sigma_r r}\|\boldsymbol{U}\|_*.$$

The proof of Lemma 4 is presented in Appendix D.6.2.

### D.5 A fourth moment bound for $\bar{\gamma}^2$

Recall from Equation (6) that the averaged measurement $\bar{\gamma}^2$ takes the form

$$\bar{\gamma}_i^2 = \frac{1}{m}\sum_{j=1}^{m}\frac{y + \eta_i^{(j)}}{\boldsymbol{a}_i^\top \boldsymbol{\Sigma}^\star \boldsymbol{a}_i} = \frac{y + \bar{\eta}_i}{\boldsymbol{a}_i^\top \boldsymbol{\Sigma}^\star \boldsymbol{a}_i}.$$

Throughout our analysis, we utilize the fact that $\bar{\gamma}^2$ has a bounded fourth moment, as characterized in the following lemma.

**Lemma 5.** *Assume $r > 8$. Then there exists a universal constant $c > 0$, such that*

$$\mathbb{E}\left(\bar{\gamma}^2\right)^4 \le c\left(\frac{y + \eta^\uparrow}{\sigma_r r}\right)^4,$$

*where $\sigma_r$ is the smallest non-zero singular value of $\boldsymbol{\Sigma}^\star$.*

The proof of Lemma 5 is presented in Appendix D.6.3. For notational simplicity of the proofs, we denote $M = c\left(\frac{y+\eta^\uparrow}{\sigma_r r}\right)^4$.

### D.6 Proofs of preliminary lemmas

In this section, we present proofs for preliminary lemmas from Appendices D.1, D.4, and D.5.

### D.6.1 Proof of Lemma 1

Using the independence of the noise $\eta$ and the sensing vector $\boldsymbol{a}$, and the assumption that $\eta$ is zero mean, we have

$$
\begin{aligned}
\mathbb{E}\left[\eta \boldsymbol{A}^{\mathsf{inv}}\right] &= \mathbb{E}\left[\frac{\eta(y+\eta)}{\boldsymbol{a}^\top \boldsymbol{\Sigma}^\star \boldsymbol{a}} \boldsymbol{a}\boldsymbol{a}^\top\right] \\
&= \mathbb{E}\left[\eta(y+\eta)\right] \cdot \mathbb{E}\left[\frac{1}{\boldsymbol{a}^\top \boldsymbol{\Sigma}^\star \boldsymbol{a}} \boldsymbol{a}\boldsymbol{a}^\top\right] \\
&= \nu_\eta^2 \, \mathbb{E}\left[\frac{1}{\boldsymbol{a}^\top \boldsymbol{\Sigma}^\star \boldsymbol{a}} \boldsymbol{a}\boldsymbol{a}^\top\right].
\end{aligned} \tag{17}
$$

The expectation in (17) is non-zero, because the random matrix $\frac{1}{\boldsymbol{a}^\top \boldsymbol{\Sigma}^\star \boldsymbol{a}} \boldsymbol{a}\boldsymbol{a}^\top$ is symmetric positive definite almost surely. Therefore, we have $\mathbb{E}\left[\eta \boldsymbol{A}^{\mathsf{inv}}\right] \neq \boldsymbol{0}$, as desired.

### D.6.2 Proof of Lemma 4

Since $\boldsymbol{\Sigma}^\star$ is symmetric positive semidefinite, it be decomposed as $\boldsymbol{Q}\boldsymbol{\Sigma}\boldsymbol{Q}^\top$, where $\boldsymbol{Q}$ is a square orthonormal matrix and $\boldsymbol{\Sigma}$ is a diagonal matrix with non-negative entries. Multiplying $\boldsymbol{a}$ by any square orthonormal matrix does not change its distribution. Therefore, without loss of generality, we assume that $\boldsymbol{\Sigma}^\star$ is diagonal with all non-negative diagonal entries. We first note that the moments of the ratios in both parts of Lemma 4 exist, because by [48, Proposition 1], for non-negative integers $p$ and $q$, the quantity $\mathbb{E}\frac{\left(\boldsymbol{a}^\top \boldsymbol{U}\boldsymbol{a}\right)^p}{\left(\boldsymbol{a}^\top \boldsymbol{\Sigma}^\star \boldsymbol{a}\right)^q}$ exists if $\frac{r}{2} > q$. Furthermore, we use the following expression from [48, Proposition 2]:

$$
\mathbb{E}\frac{\left(\boldsymbol{a}^\top \boldsymbol{U}\boldsymbol{a}\right)^p}{\left(\boldsymbol{a}^\top \boldsymbol{\Sigma}^\star \boldsymbol{a}\right)^q} = \frac{1}{\Gamma(q)} \int_0^\infty t^{q-1} \cdot |\boldsymbol{\Delta}_t| \cdot \mathbb{E}\left(\boldsymbol{a}^\top \boldsymbol{\Delta}_t \boldsymbol{U} \boldsymbol{\Delta}_t \boldsymbol{a}\right)^p \mathrm{d}t, \tag{18}
$$

where $\boldsymbol{\Delta}_t = (\boldsymbol{I}_d + 2t\boldsymbol{\Sigma}^\star)^{-1/2}$ and $|\boldsymbol{\Delta}_t|$ is the determinant of $\boldsymbol{\Delta}_t$. To characterize the determinant $|\boldsymbol{\Delta}_t|$, we note that $\boldsymbol{\Delta}_t$ is a diagonal matrix whose $d$ diagonal entries are

$$
\frac{1}{(1+2t\sigma_1)^{1/2}}, \ \ldots, \ \frac{1}{(1+2t\sigma_r)^{1/2}}, \ 1, \ \ldots, \ 1.
$$

Hence, the determinant is the product $|\boldsymbol{\Delta}_t| = \prod_{i=1}^r \frac{1}{(1+2t\sigma_i)^{1/2}}$. Furthermore, this product can be bounded as:

$$
|\boldsymbol{\Delta}_t| \leq \frac{1}{(1+2t\sigma_r)^{r/2}}. \tag{19}
$$

We now prove parts (a) and (b) separately.

**Part (a).** Using the integral expression (18) with $p = 0$ and $q = 4$, and the upper bound (19) on the determinant, we have

$$
\begin{aligned}
\mathbb{E}\left(\frac{1}{\boldsymbol{a}^\top \boldsymbol{\Sigma}^\star \boldsymbol{a}}\right)^4 &= \frac{1}{\Gamma(4)} \int_0^\infty t^3 \cdot |\boldsymbol{\Delta}_t| \, \mathrm{d}t \\
&\leq \frac{1}{\Gamma(4)} \int_0^\infty t^3 \frac{1}{(1+2t\sigma_r)^{r/2}} \, \mathrm{d}t.
\end{aligned}
$$

Denoting $s := 1 + 2t\sigma_r$, we have

$$\mathbb{E}\left(\frac{1}{\boldsymbol{a}^\top \boldsymbol{\Sigma}^\star \boldsymbol{a}}\right)^4 \leq \frac{1}{2\Gamma(4)\sigma_r} \int_1^\infty \left(\frac{s-1}{2\sigma_r}\right)^3 \frac{1}{s^{r/2}}\,\mathrm{d}s$$

$$\lesssim \frac{1}{\sigma_r^4} \int_1^\infty \frac{(s-1)^3}{s^{r/2}}\,\mathrm{d}s$$

$$= \frac{1}{\sigma_r^4} \int_1^\infty \left(\frac{s^3}{s^{r/2}} - 3\frac{s^2}{s^{r/2}} + 3\frac{s}{s^{r/2}} - \frac{1}{s^{r/2}}\right)\,\mathrm{d}s$$

$$= \frac{1}{\sigma_r^4}\left(\frac{2}{r-8} - \frac{6}{r-6} + \frac{6}{r-4} - \frac{2}{r-2}\right)$$

$$\leq \frac{c}{\sigma_r^4 r^4},$$

as desired.

**Part (b).** Using the integral expression (18) $p = q = 1$ and the upper bound (19) on the determinant, we have

$$\mathbb{E}\left(\frac{\boldsymbol{a}^\top \boldsymbol{U} \boldsymbol{a}}{\boldsymbol{a}^\top \boldsymbol{\Sigma}^\star \boldsymbol{a}}\right) = \frac{1}{\Gamma(1)} \int_0^\infty |\boldsymbol{\Delta}_t| \cdot \mathbb{E}\left[\boldsymbol{a}^\top \boldsymbol{\Delta}_t \boldsymbol{U} \boldsymbol{\Delta}_t \boldsymbol{a}\right]\,\mathrm{d}t$$

$$\leq \frac{1}{\Gamma(1)} \int_0^\infty \frac{1}{(1+2t\sigma_r)^{r/2}} \mathbb{E}\left[\boldsymbol{a}^\top \boldsymbol{\Delta}_t \boldsymbol{U} \boldsymbol{\Delta}_t \boldsymbol{a}\right]\,\mathrm{d}t. \qquad (20)$$

We now bound the expectation term in (20). Note that for $\boldsymbol{a} \sim \mathcal{N}(\boldsymbol{0}, \boldsymbol{I}_d)$, we have $\mathbb{E}\left[\boldsymbol{a}^\top \boldsymbol{B} \boldsymbol{a}\right] = \mathrm{tr}\,(\boldsymbol{B})$ for any symmetric matrix $\boldsymbol{B}$. Therefore, we have

$$\mathbb{E}\left[\boldsymbol{a}^\top \boldsymbol{\Delta}_t \boldsymbol{U} \boldsymbol{\Delta}_t \boldsymbol{a}\right] = \mathrm{tr}\,(\boldsymbol{\Delta}_t \boldsymbol{U} \boldsymbol{\Delta}_t)$$

$$\overset{(i)}{\leq} \|\boldsymbol{\Delta}_t \boldsymbol{U} \boldsymbol{\Delta}_t\|_*$$

$$\overset{(ii)}{\leq} \|\boldsymbol{U}\|_*, \qquad (21)$$

where (i) the fact that $\mathrm{tr}\,(\boldsymbol{B}) \leq \|\boldsymbol{B}\|_*$ for any symmetric matrix $\boldsymbol{B}$. Furthermore, (ii) follows from Hölder's inequality for Schatten-$p$ norms, where we have that $\|\boldsymbol{\Delta}_t \boldsymbol{U} \boldsymbol{\Delta}_t\|_* \leq \|\boldsymbol{\Delta}_t\|_{\mathrm{op}}^2 \cdot \|\boldsymbol{U}\|_*$. Because $\boldsymbol{\Delta}_t$ is diagonal and the entries of $\boldsymbol{\Delta}_t$ are bounded between $0$ and $1$, we bound the operator norm as $\|\boldsymbol{\Delta}_t\|_{\mathrm{op}} \leq 1$. Substituting (21) to (20), we obtain

$$\mathbb{E}\left(\frac{\boldsymbol{a}^\top \boldsymbol{U} \boldsymbol{a}}{\boldsymbol{a}^\top \boldsymbol{\Sigma}^\star \boldsymbol{a}}\right) \leq \|\boldsymbol{U}\|_* \cdot \int_0^\infty \frac{1}{(1+2t\sigma_r)^{r/2}}\,\mathrm{d}t$$

$$\lesssim \frac{1}{\sigma_r r} \cdot \|\boldsymbol{U}\|_*,$$

as desired.

### D.6.3 Proof of Lemma 5

By the assumption that the noise is upper bounded by $\eta^\uparrow$, we have $y + \bar{\eta} \leq y + \eta^\uparrow$. Therefore, we have

$$\mathbb{E}\left(\bar{\gamma}^2\right)^4 = \mathbb{E}\left(\frac{y + \bar{\eta}}{\boldsymbol{a}^\top \boldsymbol{\Sigma}^\star \boldsymbol{a}}\right)^4$$

$$\leq (y + \eta^\uparrow)^4 \cdot \mathbb{E}\left(\frac{1}{\boldsymbol{a}^\top \boldsymbol{\Sigma}^\star \boldsymbol{a}}\right)^4$$

$$\overset{(i)}{\lesssim} \left(\frac{1}{\sigma_r r}\right)^4,$$

where step (i) applies part (a) of Lemma 4.

## E   Proof of Theorem 1

Recall that we assume we collect $N$ measurements under the inverted measurements sensing model presented in Algorithm 1 with standard Gaussian sensing vectors and bounded noise, mean-zero noise (Assumption 1).

We first introduce a restricted strong convexity (RSC) condition that our proof relies on. Since the matrix $\boldsymbol{\Sigma}^\star$ is assumed to be symmetric positive semidefinite matrices and of rank $r$, we follow [24] and consider a restricted set on which we analyze the behavior of the sensing matrices $\widetilde{\boldsymbol{A}}_i$. We call this set the "error set", defined by:

$$\mathcal{E} = \left\{\boldsymbol{U} \in \mathbb{S}^{d \times d} : \|\boldsymbol{U}\|_* \leq 4\sqrt{2r}\|\boldsymbol{U}\|_F\right\}, \tag{22}$$

where recall that $\mathbb{S}^{d \times d}$ denotes the set of symmetric $d \times d$ matrices. We say that our shrunken sensing matrices $\{\widetilde{\boldsymbol{A}}_i\}_{i=1}^n$ satisfy a restricted strong convexity (RSC) condition over the error set $\mathcal{E}$, if there exists some positive constant $\kappa > 0$ such that

$$\frac{1}{n}\sum_{i=1}^n \langle \widetilde{\boldsymbol{A}}_i, \boldsymbol{U}\rangle^2 \geq \kappa \|\boldsymbol{U}\|_F^2 \qquad \text{for all } \boldsymbol{U} \in \mathcal{E}. \tag{23}$$

The following proposition shows that the estimation error, when the sensing matrices satisfy the RSC condition and the regularization parameter is sufficiently large.

**Proposition 1** ([39, Theorem 1] with $q = 0$). *Suppose that $\boldsymbol{\Sigma}^\star$ has rank $r$ and the shrunken sensing matrices satisfy the restricted strong convexity condition* (23) *with positive constant $\kappa > 0$. Then if the regularization parameter satisfies*

$$\lambda_n \geq 2\left\|\frac{1}{n}\sum_{i=1}^n y\widetilde{\boldsymbol{A}}_i - \frac{1}{n}\sum_{i=1}^n \langle \widetilde{\boldsymbol{A}}_i, \boldsymbol{\Sigma}^\star\rangle \widetilde{\boldsymbol{A}}_i\right\|_{\text{op}}, \tag{24}$$

*any optimal solution $\widehat{\boldsymbol{\Sigma}}$ of the optimization program* (8) *satisfies*

$$\|\widehat{\boldsymbol{\Sigma}} - \boldsymbol{\Sigma}^\star\|_F \leq \frac{32\sqrt{r}\lambda_n}{\kappa}.$$

This theorem is a special case of Theorem 1 in [39], which is in turn adapted from Theorem 1 in [24] (see [24] or [39] for the proof). Proposition 1 is a deterministic and nonasymptotic result and provides a roadmap for proving our desired upper bound. First, we show that the operator norm (24) can be upper bounded with high probability, allowing us to set the regularization parameter $\lambda_n$ accordingly. Second, we show that the RSC condition (23) is satisfied with high probability. We begin by bounding the operator norm (24) in the following proposition.

**Proposition 2.** *Let $y^\uparrow = y + \eta^\uparrow$. Suppose that $\boldsymbol{\Sigma}^\star$ has rank $r$, with $r > 8$. Then there exists a positive absolute constant $C_1$ such that*

$$\left\|\frac{1}{n}\sum_{i=1}^n y\widetilde{\boldsymbol{A}}_i - \frac{1}{n}\sum_{i=1}^n \langle \widetilde{\boldsymbol{A}}_i, \boldsymbol{\Sigma}^\star\rangle \widetilde{\boldsymbol{A}}_i\right\|_{\text{op}} \leq C_1\left[y^\uparrow\left(\frac{y^\uparrow}{\sigma_r r}\sqrt{\frac{d}{n}} + \frac{d}{n}\tau + \left(\frac{y^\uparrow}{\sigma_r r}\right)^2 \frac{1}{\tau}\right) + \frac{1}{\sigma_r r}\frac{\nu_\eta^2}{m}\right] \tag{25}$$

*with probability at least $1 - 4\exp(-d)$.*

The proof of Proposition 2 is provided in Appendix E.1. Next, we show that the RSC condition (23) is satisfied with high probability, as is done in the following proposition.

**Proposition 3.** *Let $\mu_y$ be the median of $y + \bar{\eta}$. Suppose that the truncation threshold $\tau$ satisfies $\tau \geq \frac{\mu_y}{\text{tr}(\Sigma^\star)}$. Then there exist positive absolute constants $\kappa_{\mathcal{L}}$, $c$, and $C$ such that if the number of sensing vectors satisfy*

$$n \geq Crd$$

*then we have*

$$\frac{1}{n} \sum_{i=1}^{n} \langle \widetilde{A}_i, U \rangle^2 \geq \kappa_{\mathcal{L}} \left( \frac{\mu_y}{\text{tr}(\Sigma^\star)} \right)^2 \|U\|_F^2 \tag{26}$$

*simultaneously for all matrices $U \in \mathcal{E}$ with probability at least $1 - \exp(-cn)$, where $\mathcal{E}$ is the error set defined in Equation (22).*

The proof of Proposition 3 is provided in Appendix E.2. We now utilize the results of Propositions 1, 2 and 3 to derive our final error bound. By Proposition 2, the operator norm (24) can be upper bounded with high probability. We set the regularization parameter $\lambda_n$ to satisfy

$$\lambda_n \geq C_1 \left[ y^\uparrow \left( \frac{y^\uparrow}{\sigma_r r} \sqrt{\frac{d}{n}} + \frac{d}{n} \tau + \left( \frac{y^\uparrow}{\sigma_r r} \right)^2 \frac{1}{\tau} \right) + \frac{1}{\sigma_r r} \frac{\nu_\eta^2}{m} \right],$$

where $C_1$ is the constant in Proposition 2. Furthermore, by Proposition 3, we have that there exist universal constant $C_2 > 0$ such that if the number of sensing vectors satisfies $n \geq C_2 rd$, the RSC condition (23) holds for constant $\kappa = \kappa_{\mathcal{L}} \left( \frac{\mu_y}{\text{tr}(\Sigma^\star)} \right)^2$ with high probability. Taking a union bound, we have that Proposition 2 and Proposition 3 hold simultaneously with probability at least $1 - 4\exp(-d) - \exp(-cn)$. Invoking Proposition 1, we have

$$\|\widehat{\Sigma} - \Sigma^\star\|_F \leq 32\sqrt{r} \cdot \frac{\lambda_n}{\kappa_{\mathcal{L}} \left( \frac{\mu_y}{\text{tr}(\Sigma^\star)} \right)^2}$$

$$\lesssim \left( \frac{\text{tr}(\Sigma^\star)}{\mu_y} \right)^2 \sqrt{r} \lambda_n$$

with probability at least $1 - 4\exp(-d) - \exp(-cn)$, as desired.

### E.1 Proof of Proposition 2

In the proof, we decompose the operator norm $\left\| \frac{1}{n} \sum_{i=1}^{n} y \widetilde{A}_i - \frac{1}{n} \sum_{i=1}^{n} \langle \widetilde{A}_i, \Sigma^\star \rangle \widetilde{A}_i \right\|_{\text{op}}$ from (25) into individual terms and bound them separately. Recall the definitions of random matrices $\bar{A}$ from Equation 15 and $\widetilde{A}$ from Equation 16.

**Step 1: decompose the error into five terms.** We begin by adding and subtracting multiple quantities as follows:

$$\frac{1}{n} \sum_{i=1}^{n} y \widetilde{A}_i - \frac{1}{n} \sum_{i=1}^{n} \langle \widetilde{A}_i, \Sigma^\star \rangle \widetilde{A}_i = \frac{1}{n} \sum_{i=1}^{n} y \widetilde{A}_i - \mathbb{E}\left[ y \widetilde{A} \right] + \mathbb{E}\left[ y \widetilde{A} \right] - \mathbb{E}\left[ y \bar{A} \right]$$

$$+ \mathbb{E}\left[ y \bar{A} \right] - \mathbb{E}\left[ \langle \widetilde{A}, \Sigma^\star \rangle \widetilde{A} \right] + \mathbb{E}\left[ \langle \widetilde{A}, \Sigma^\star \rangle \widetilde{A} \right] - \frac{1}{n} \sum_{i=1}^{n} \langle \widetilde{A}_i, \Sigma^\star \rangle \widetilde{A}_i$$

$$\overset{(i)}{=} \frac{1}{n} \sum_{i=1}^{n} y \widetilde{A}_i - \mathbb{E}\left[ y \widetilde{A} \right] + \mathbb{E}\left[ y \widetilde{A} \right] - \mathbb{E}\left[ y \bar{A} \right]$$

$$+ \mathbb{E}\left[ \langle \bar{A}, \Sigma^\star \rangle \bar{A} \right] - \mathbb{E}\left[ \langle \widetilde{A}, \Sigma^\star \rangle \widetilde{A} \right] - \mathbb{E}\left[ \bar{\eta} \bar{A} \right]$$

$$+ \mathbb{E}\left[ \langle \widetilde{A}, \Sigma^\star \rangle \widetilde{A} \right] - \frac{1}{n} \sum_{i=1}^{n} \langle \widetilde{A}_i, \Sigma^\star \rangle \widetilde{A}_i, \tag{27}$$

where step (i) is true by substituting $y = \langle \bar{A}, \Sigma^\star \rangle - \bar{\eta}$ to the term of $\mathbb{E}\left[y\bar{A}\right]$, and the fact that the noise term $\bar{\eta}$ is zero-mean. By triangle inequality, we group the terms in (27) and bound the operator norm by

$$\left\| \frac{1}{n}\sum_{i=1}^n y\widetilde{A}_i - \frac{1}{n}\sum_{i=1}^n \langle \widetilde{A}_i, \Sigma^\star \rangle \widetilde{A}_i \right\|_{\text{op}} \leq \underbrace{y \left\| \frac{1}{n}\sum_{i=1}^n \widetilde{A}_i - \mathbb{E}\left[\widetilde{A}\right] \right\|_{\text{op}}}_{\text{Term 1}}$$

$$+ \underbrace{y \left\| \mathbb{E}\left[\widetilde{A}\right] - \mathbb{E}\left[\bar{A}\right] \right\|_{\text{op}}}_{\text{Term 2}}$$

$$+ \underbrace{\left\| \mathbb{E}\left[\langle \bar{A}, \Sigma^\star \rangle \bar{A}\right] - \mathbb{E}\left[\langle \widetilde{A}, \Sigma^\star \rangle \widetilde{A}\right] \right\|_{\text{op}}}_{\text{Term 3}}$$

$$+ \underbrace{\left\| \mathbb{E}\left[\langle \widetilde{A}, \Sigma^\star \rangle \widetilde{A}\right] - \frac{1}{n}\sum_{i=1}^n \langle \widetilde{A}_i, \Sigma^\star \rangle \widetilde{A}_i \right\|_{\text{op}}}_{\text{Term 4}}$$

$$+ \underbrace{\left\| \mathbb{E}\left[\bar{\eta}\bar{A}\right] \right\|_{\text{op}}}_{\text{Term 5}}. \tag{28}$$

In the remaining proof, we bound the five terms in (28) individually. We first discuss the nature of these five terms.

- **Terms 1 and 4:** These two terms characterize the difference between the empirical mean of quantities involving $\widetilde{A}$ and their true expectation (see Lemma 6 and Lemma 9). In the proof, we show that the empirical mean concentrates around the expectation with high probability, as a function of the number of sensing vectors $n$.

- **Terms 2 and 3:** These two terms characterize the difference in expectation introduced by truncating $\bar{A}$ to $\widetilde{A}$ (see Lemma 7 and Lemma 8). Hence, these two terms characterize biases that arise from truncation. They diminish as $\tau \to \infty$, because setting $\tau$ to $\infty$ is equivalent to no thresholding, and hence $\widetilde{A}$ becomes identical to $\bar{A}$. Since expectations are considered, these two terms depend on the threshold $\tau$, but not the number of sensing vectors $n$ or the averaging parameter $m$.

- **Term 5:** Term 5 is a bias term that arises from the fact that the mean of the noise $\eta$ conditioned on sensing matrix $\bar{A}$ is non-zero. We show that this bias scales like $\frac{1}{m}$ (see Lemma 10) in terms of the averaging parameter $m$.

Putting these terms together, Terms 1 and 4 depend on $n$, Terms 2 and 3 depend on $\tau$, and Term 5 depends on $m$. In Corollary 1, we set the values of $\tau$, $n$ and $m$ to balance these terms.

**Step 2: bound the five terms individually.** In what follows, we provide five lemmas to bound each of the five terms individually. In the proofs of the five lemmas, we rely on an upper bound on the fourth moment of the $m$-sample averaged measurements $\bar{\gamma}^2$. As shown in Lemma 5 in Appendix D.5, for some absolute constant $c$, this fourth moment can be upper bounded by a quantity that we denote $M$:

$$\mathbb{E}[(\bar{\gamma}^2)^4] \leq M = c\left(\frac{y^\uparrow}{\sigma_r r}\right)^4. \tag{29}$$

We also rely heavily on the following truncation properties relating the $m$-sample averaged measurements $\bar{\gamma}^2$ and truncated measurements $\tilde{\gamma}^2$:

$$\tilde{\gamma}_i^2 \leq \tau \tag{TP1}$$

$$\tilde{\gamma}_i^2 \leq \bar{\gamma}_i^2 \tag{TP2}$$

$$\bar{\gamma}_i^2 - \tilde{\gamma}_i^2 = (\bar{\gamma}_i^2 - \tilde{\gamma}_i^2) \cdot \mathbf{1}\{\bar{\gamma}_i^2 \geq \tau\}. \tag{TP3}$$

The following lemma provides a bound for Term 1.

**Lemma 6.** *Let* $\widetilde{\boldsymbol{A}}_1, \ldots, \widetilde{\boldsymbol{A}}_n$ *be i.i.d copies of a random matrix* $\widetilde{\boldsymbol{A}}$ *as defined in Equation* (16)*. There exists an absolute constant* $c > 0$ *such that for any* $t > 0$*, we have*

$$\left\| \frac{1}{n} \sum_{i=1}^{n} \widetilde{\boldsymbol{A}}_i - \mathbb{E}\left[\widetilde{\boldsymbol{A}}_i\right] \right\|_{\mathrm{op}} \leq c \left( \sqrt{\frac{M^{1/2}t}{n}} + \frac{\tau t}{n} \right)$$

*with probability at least* $1 - 2 \cdot 9^d \cdot \exp(-t)$.

The proof of Lemma 6 is provided in Appendix E.1.1. The next lemma provides an upper bound for Term 2.

**Lemma 7.** *Let* $\bar{\boldsymbol{A}}$ *and* $\widetilde{\boldsymbol{A}}$ *be the random matrices defined in Equation* (15) *and Equation* (16)*, respectively. Then there exists an absolute constant* $c > 0$ *such that*

$$\left\| \mathbb{E}\left[\widetilde{\boldsymbol{A}}\right] - \mathbb{E}\left[\bar{\boldsymbol{A}}\right] \right\|_{\mathrm{op}} \leq \frac{cM^{1/2}}{\tau}.$$

The proof of Lemma 7 is provided in Appendix E.1.2. The following lemma provides an upper bound for Term 3. Recall that the quantity $y^{\uparrow}$ denotes $y + \eta^{\uparrow}$.

**Lemma 8.** *Let* $\bar{\boldsymbol{A}}$ *and* $\widetilde{\boldsymbol{A}}$ *be the random matrices defined in Equation* (15) *and Equation* (16)*, respectively. Then there exists an absolute constant* $c > 0$ *such that*

$$\left\| \mathbb{E}\left[\langle\bar{\boldsymbol{A}}, \boldsymbol{\Sigma}^{\star}\rangle\bar{\boldsymbol{A}}\right] - \mathbb{E}\left[\langle\widetilde{\boldsymbol{A}}, \boldsymbol{\Sigma}^{\star}\rangle\widetilde{\boldsymbol{A}}\right] \right\|_{\mathrm{op}} \leq \frac{c\, y^{\uparrow}M^{1/2}}{\tau}.$$

The proof of Lemma 8 is provided in Appendix E.1.3. The following lemma provides an upper bound for Term 4.

**Lemma 9.** *Let* $\widetilde{\boldsymbol{A}}_1, \ldots, \widetilde{\boldsymbol{A}}_n$ *be i.i.d copies of a random matrix* $\widetilde{\boldsymbol{A}}$ *defined in Equation* (16)*. There exists an absolute constant* $c > 0$ *such that for any* $t > 0$*, we have*

$$\left\| \mathbb{E}\left[\langle\widetilde{\boldsymbol{A}}, \boldsymbol{\Sigma}^{\star}\rangle\widetilde{\boldsymbol{A}}\right] - \frac{1}{n} \sum_{i=1}^{n} \langle\widetilde{\boldsymbol{A}}_i, \boldsymbol{\Sigma}^{\star}\rangle\widetilde{\boldsymbol{A}}_i \right\|_{\mathrm{op}} \leq c\, y^{\uparrow} \left( \sqrt{\frac{M^{1/2}t}{n}} + \frac{\tau t}{n} \right)$$

*with probability at least* $1 - 2 \cdot 9^d \cdot \exp(-t)$.

The proof of Lemma 9 is provided in Appendix E.1.4. We note that Terms 2 and 3 are bias that result from shrinkage, but crucially are inversely dependent on the shrinkage threshold $\tau$. This fact allows us to set $\tau$ so that the order of Terms 2 and 3 match the order of Terms 1 and 4.

The final lemma bounds Term 5, which is a bias that arises from the dependence of the sensing matrix $\bar{\boldsymbol{A}}$ on the noise $\eta$.

**Lemma 10.** *Let* $\bar{\boldsymbol{A}}$ *be the random matrix defined in Equation* (15)*. Suppose that* $\boldsymbol{\Sigma}^{\star}$ *has rank* $r$ *with* $r > 2$*. Then there exists an absolute constant* $c > 0$ *such that*

$$\mathbb{E}\left[\left\|\bar{\eta}\bar{\boldsymbol{A}}\right\|_{\mathrm{op}}\right] \leq \frac{c}{\sigma_r r} \frac{\nu_\eta^2}{m}.$$

The proof of Lemma 10 is provided in Appendix E.1.5. We note that the bias scales with the variance of the $m$-sample averaged noise $\bar{\eta}$, which scales inversely with $m$.

**Step 3: combine the five terms.** We set $t = (\log 9 + 1)d$. Substituting the bounds from Lemmas 6–10 back to (28) and taking a union bound, we have that with probability at least $1 - 4\exp(-d)$,

$$\left\| \frac{1}{n} \sum_{i=1}^{n} y\widetilde{\boldsymbol{A}}_i - \frac{1}{n} \sum_{i=1}^{n} \langle\widetilde{\boldsymbol{A}}_i, \boldsymbol{\Sigma}^{\star}\rangle\widetilde{\boldsymbol{A}}_i \right\|_{\mathrm{op}} \lesssim \left(y^{\uparrow} + 1\right) \left( \sqrt{\frac{M^{1/2}d}{n}} + \frac{d}{n}\tau + \frac{M^{1/2}}{\tau} \right) + \frac{1}{\sigma_r r} \frac{\nu_\eta^2}{m}$$

$$\overset{(i)}{\lesssim} y^{\uparrow} \left( \frac{y^{\uparrow}}{\sigma_r r} \sqrt{\frac{d}{n}} + \frac{d}{n}\tau + \left(\frac{y^{\uparrow}}{\sigma_r r}\right)^2 \frac{1}{\tau} \right) + \frac{1}{\sigma_r r} \frac{\nu_\eta^2}{m},$$

where step (i) is true by substituting in the expression (29) for $M$.

### E.1.1 Proof of Lemma 6.

Let $\mathcal{A}_{\frac{1}{4}} \subseteq \mathcal{S}^{d-1}$ be a $\frac{1}{4}$-covering of the $d$-dimensional unit sphere $\mathcal{S}^{d-1} := \{x \in \mathbb{R}^d : \|x\|_2 = 1\}$. By a covering argument [46, Exercise 4.4.3], for any symmetric matrix $\boldsymbol{U} \in \mathbb{S}^{d \times d}$, its operator norm is bounded by $\|\boldsymbol{U}\|_{\text{op}} \le 2 \sup_{\boldsymbol{v} \in \mathcal{A}_{\frac{1}{4}}} |\boldsymbol{v}^\top \boldsymbol{U} \boldsymbol{v}|$. Hence, we have

$$
\left\| \frac{1}{n} \sum_{i=1}^n \widetilde{\boldsymbol{A}}_i - \mathbb{E}\left[\widetilde{\boldsymbol{A}}\right] \right\|_{\text{op}} \le 2 \sup_{\boldsymbol{v} \in \mathcal{A}_{\frac{1}{4}}} \left| \boldsymbol{v}^\top \left( \frac{1}{n} \sum_{i=1}^n \widetilde{\boldsymbol{A}}_i - \mathbb{E}\left[\widetilde{\boldsymbol{A}}\right] \right) \boldsymbol{v} \right|
$$
$$
= 2 \sup_{\boldsymbol{v} \in \mathcal{A}_{\frac{1}{4}}} \left| \frac{1}{n} \sum_{i=1}^n \boldsymbol{v}^\top \widetilde{\boldsymbol{A}}_i \boldsymbol{v} - \mathbb{E}\left[\boldsymbol{v}^\top \widetilde{\boldsymbol{A}} \boldsymbol{v}\right] \right|. \tag{30}
$$

We invoke Bernstein's inequality. We first show that the Bernstein condition holds. Namely, we show that for each integer $p \ge 2$, we have that for any unit vector $\boldsymbol{v} \in \mathbb{R}^d$,

$$
\mathbb{E}\left| \boldsymbol{v}^\top \widetilde{\boldsymbol{A}} \boldsymbol{v} \right|^p \le \frac{p!}{2} u_1 u_2^{p-2}, \tag{31}
$$

where $u_1 = c_1 M^{\frac{1}{2}}$ and $u_2 = c_2 \tau$ for some universal positive constants $c_1$ and $c_2$. Given the Bernstein condition (31), we then apply Bernstein's inequality to bound (30).

**Proving the Bernstein condition** (31).   We fix any unit vector $\boldsymbol{v} \in \mathbb{R}^d$. Since $\widetilde{\boldsymbol{A}} = \widetilde{\gamma}^2 \boldsymbol{a} \boldsymbol{a}^\top$, we have $\boldsymbol{v}^\top \widetilde{\boldsymbol{A}} \boldsymbol{v} = \widetilde{\gamma}^2 (\boldsymbol{v}^\top \boldsymbol{a})^2$. Recall that the random vector $\boldsymbol{a}$ is distributed as $\boldsymbol{a} \sim \mathcal{N}(\boldsymbol{0}, \boldsymbol{I}_d)$. Since $\boldsymbol{v}$ is a unit vector, it follows that $\boldsymbol{v}^\top \boldsymbol{a} \sim \mathcal{N}(0, 1)$. Denote by $G \sim \mathcal{N}(0, 1)$ a standard normal random variable. For any integer $p \ge 2$, we have

$$
\mathbb{E}\left| \boldsymbol{v}^\top \widetilde{\boldsymbol{A}} \boldsymbol{v} \right|^p = \mathbb{E}\left( \widetilde{\gamma}^2 G^2 \right)^p \overset{\text{(i)}}{\le} \tau^{p-2} \mathbb{E}\left[ \left(\widetilde{\gamma}^2\right)^2 G^{2p} \right]
$$
$$
\overset{\text{(ii)}}{\le} \tau^{p-2} \cdot \mathbb{E}\left[ \left(\bar{\gamma}^2\right)^2 G^{2p} \right]
$$
$$
\overset{\text{(iii)}}{\le} \tau^{p-2} \left( \mathbb{E}\left[ \left(\bar{\gamma}^2\right)^4 \right] \cdot \mathbb{E}\left[ G^{4p} \right] \right)^{1/2}
$$
$$
\overset{\text{(iv)}}{\le} \tau^{p-2} \left( M \cdot \mathbb{E}\left[ G^{4p} \right] \right)^{1/2}, \tag{32}
$$

where steps (i) and (ii) follow from (TP1) and (TP2), respectively; step (iii) follows from Cauchy–Schwarz inequality; and step (iv) follows upper bounding the fourth moment of $\bar{\gamma}^2$ with the quantity $M$ from Equation (29).

Note that since $G$ is standard normal, by definition $G^2$ follows a Chi-Square distribution with 1 degree of freedom, and hence sub-exponential. By Lemma 2 in Appendix D.2, there exists some constant $c > 0$ such that we have $\left( \mathbb{E}\left[ (G^2)^p \right] \right)^{1/p} \le cp$ for all $p \ge 1$. Hence, we have $\left( \mathbb{E}\left[ G^{4p} \right] \right)^{1/2p} \le 2cp$ and

$$
\left( \mathbb{E}\left[ G^{4p} \right] \right)^{1/2} \le (2cp)^p = \left( \frac{p}{e} \right)^p \cdot (2ec)^p
$$
$$
\overset{\text{(i)}}{<} p! \cdot (2ec)^p \tag{33}
$$

where step (i) is true by Stirling's inequality that for all $p \ge 1$,

$$
p! > \sqrt{2\pi p} \left( \frac{p}{e} \right)^p e^{\frac{1}{12p+1}} > \left( \frac{p}{e} \right)^p.
$$

Substituting (33) back to (32) and rearranging terms completes the proof of the Bernstein condition (31).

**Applying Bernstein's inequality to bound** (30).   By Bernstein's inequality (see Lemma 3), given condition (31), we have that for any unit vector $\boldsymbol{v} \in \mathbb{R}^d$ and any $t > 0$,

$$
\mathbb{P}\left( \left| \frac{1}{n} \sum_{i=1}^n \boldsymbol{v}^\top \widetilde{\boldsymbol{A}}_i \boldsymbol{v} - \mathbb{E}\left[\boldsymbol{v}^\top \widetilde{\boldsymbol{A}} \boldsymbol{v}\right] \right| \ge 2 \left( \sqrt{\frac{c_1 M^{1/2} t}{n}} + \frac{c_2 \tau t}{n} \right) \right) \le 2 \exp(-t). \tag{34}
$$

By [46, Corollary 4.2.13], the cardinality of the covering set $\mathcal{A}_{\frac{1}{4}}$ is bounded above by $9^d$. Therefore, taking a union bound on (34), we have

$$\mathbb{P}\left(\sup_{\boldsymbol{v}\in\mathcal{A}_{\frac{1}{4}}}\left|\frac{1}{n}\sum_{i=1}^{n}\boldsymbol{v}^\top\widetilde{\boldsymbol{A}}_i\boldsymbol{v} - \mathbb{E}\left[\boldsymbol{v}^\top\widetilde{\boldsymbol{A}}\boldsymbol{v}\right]\right| \geq 2\left(\sqrt{\frac{c_1 M^{1/2} t}{n}} + \frac{c_2 \tau t}{n}\right)\right) \leq 2\cdot 9^d\cdot\exp\left(-t\right). \quad (35)$$

Substituting in (30) to (35), for any $t > 0$, we have

$$\mathbb{P}\left(\left\|\frac{1}{n}\sum_{i=1}^{n}\widetilde{\boldsymbol{A}}_i - \mathbb{E}\left[\widetilde{\boldsymbol{A}}\right]\right\|_{\text{op}} \lesssim \sqrt{\frac{M^{1/2} t}{n}} + \frac{\tau t}{n}\right) \geq 1 - 2\cdot 9^d\cdot\exp(-t),$$

as desired.

### E.1.2 Proof of Lemma 7

By definition of the operator norm, we have

$$\left\|\mathbb{E}\left[\widetilde{\boldsymbol{A}}\right] - \mathbb{E}\left[\bar{\boldsymbol{A}}\right]\right\|_{\text{op}} = \sup_{\boldsymbol{v}\in\mathcal{S}^{d-1}}\left|\boldsymbol{v}^\top\left(\mathbb{E}\left[\bar{\boldsymbol{A}}\right] - \mathbb{E}\left[\widetilde{\boldsymbol{A}}\right]\right)\boldsymbol{v}\right|.$$

We fix any $\boldsymbol{v}\in\mathcal{S}^{d-1}$, and bound $\boldsymbol{v}^\top\left(\mathbb{E}\left[\bar{\boldsymbol{A}}\right] - \mathbb{E}\left[\widetilde{\boldsymbol{A}}\right]\right)\boldsymbol{v}$. Similar to the proof of Lemma 6, we note that $\boldsymbol{v}^\top\boldsymbol{a}\sim\mathcal{N}(0,1)$ and denote the random variable $G\sim\mathcal{N}(0,1)$. Substituting in the expression for sensing matrices $\bar{\boldsymbol{A}}$ and $\widetilde{\boldsymbol{A}}$, we have

$$\left|\boldsymbol{v}^\top\left(\mathbb{E}\left[\bar{\boldsymbol{A}}\right] - \mathbb{E}\left[\widetilde{\boldsymbol{A}}\right]\right)\boldsymbol{v}\right| = \left|\boldsymbol{v}^\top\mathbb{E}\left[\bar{\gamma}^2\boldsymbol{a}\boldsymbol{a}^\top - \widetilde{\gamma}^2\boldsymbol{a}\boldsymbol{a}^\top\right]\boldsymbol{v}\right|$$

$$\stackrel{\text{(i)}}{=}\mathbb{E}\left[\left(\bar{\gamma}^2 - \widetilde{\gamma}^2\right)G^2\right]$$

$$\stackrel{\text{(ii)}}{=}\mathbb{E}\left[\left(\bar{\gamma}^2 - \widetilde{\gamma}^2\right)G^2\cdot\mathbb{1}\{\bar{\gamma}^2\geq\tau\}\right]$$

$$\leq\mathbb{E}\left[\bar{\gamma}^2 G^2\cdot\mathbb{1}\{\bar{\gamma}^2\geq\tau\}\right]$$

$$\stackrel{\text{(iii)}}{\leq}\left(\mathbb{E}\left[(\bar{\gamma}^2 G^2)^2\right]\cdot\mathbb{E}\left[\mathbb{1}\{\bar{\gamma}^2\geq\tau\}\right]\right)^{1/2}$$

$$\stackrel{\text{(iv)}}{\leq}\left(\mathbb{E}\left[|\bar{\gamma}^2|^4\right]\cdot\mathbb{E}\left[|G^2|^4\right]\right)^{1/4}\left(\mathbb{P}\left(\bar{\gamma}^2\geq\tau\right)\right)^{1/2}, \quad (36)$$

where where step (i) is true because $\bar{\gamma}^2\geq\widetilde{\gamma}^2$ from to (TP2), step (ii) is true due to (TP3), and steps (iii) and (iv) follow from Cauchy–Schwarz inequality. We proceed by bounding each of the terms in (36) separately. First, we can upper bound the fourth moment $\mathbb{E}\left[|\bar{\gamma}^2|^4\right]$ by the quantity $M$ from Equation (29). Second, $G^2$ is a sub-exponential random variable. By Lemma 2 in Appendix D.2, we have that $\mathbb{E}\left[|G^2|^4\right]^{1/4}\leq c$ for some constant $c$. It remains to bound the term $\left(\mathbb{P}\left(\bar{\gamma}^2\geq\tau\right)\right)^{1/2}$. We have

$$\mathbb{P}\left(\bar{\gamma}^2\geq\tau\right)\stackrel{\text{(i)}}{\leq}\frac{\mathbb{E}\,|\bar{\gamma}^2|^2}{\tau^2}$$

$$\stackrel{\text{(ii)}}{\leq}\frac{\left(\mathbb{E}\,|\bar{\gamma}^2|^4\right)^{1/2}}{\tau^2}$$

$$\stackrel{\text{(iii)}}{\leq}\frac{M^{1/2}}{\tau^2},$$

where step (i) follows from Markov's inequality, step (ii) follows from Cauchy–Schwarz inequality, and step (iii) follows from the fourth moment bound on the averaged scaling $\bar{\gamma}^2$. Putting everything together back to (36), we have

$$\left|\boldsymbol{v}^\top\left(\mathbb{E}\left[\bar{\boldsymbol{A}}\right] - \mathbb{E}\left[\widetilde{\boldsymbol{A}}\right]\right)\boldsymbol{v}\right|\lesssim\frac{M^{1/2}}{\tau}$$

for any vector $\boldsymbol{v}\in\mathcal{S}^{d-1}$. Therefore,

$$\left\|\mathbb{E}\left[\widetilde{\boldsymbol{A}}\right] - \mathbb{E}\left[\bar{\boldsymbol{A}}\right]\right\|_{\text{op}}\lesssim\frac{M^{1/2}}{\tau},$$

as desired.

### E.1.3 Proof of Lemma 8

Substituting in the definitions $\bar{A} = \bar{\gamma}^2 a a^\top$ and $\widetilde{A} = \widetilde{\gamma}^2 a a^\top$, we have

$$\langle \bar{A}, \Sigma^\star \rangle \bar{A} - \langle \widetilde{A}, \Sigma^\star \rangle \widetilde{A} = \left( \bar{\gamma}^4 - \widetilde{\gamma}^4 \right) \left( a^\top \Sigma^\star a \right) a a^\top.$$

Therefore, our goal is to bound the operator norm

$$\left\| \left( \bar{\gamma}^4 - \widetilde{\gamma}^4 \right) \left( a^\top \Sigma^\star a \right) a a^\top \right\|_{\mathrm{op}} = \sup_{v \in \mathcal{S}^{d-1}} \left| v^T \left( \bar{\gamma}^4 - \widetilde{\gamma}^4 \right) \left( a^\top \Sigma^\star a \right) a a^\top v \right|.$$

Similar to the proof of Lemma 7, we fix any vector $v \in \mathcal{S}^{d-1}$. Again, note that $v^\top a \sim \mathcal{N}(0,1)$ and denote $G \sim \mathcal{N}(0,1)$. We have

$$
\begin{aligned}
\left| v^\top \mathbb{E} \left[ \left( \bar{\gamma}^4 - \widetilde{\gamma}^4 \right) \left( a^\top \Sigma^\star a \right) a a^\top \right] v \right| &\overset{\text{(i)}}{=} \mathbb{E} \left[ \left( \bar{\gamma}^4 - \widetilde{\gamma}^4 \right) \left( a^\top \Sigma^\star a \right) G^2 \right] \\
&= \mathbb{E} \left[ \left( \bar{\gamma}^2 + \widetilde{\gamma}^2 \right) \left( \bar{\gamma}^2 - \widetilde{\gamma}^2 \right) \left( a^\top \Sigma^\star a \right) G^2 \right] \\
&\overset{\text{(ii)}}{\leq} \mathbb{E} \left[ 2 \bar{\gamma}^2 \left( \bar{\gamma}^2 - \widetilde{\gamma}^2 \right) \left( a^\top \Sigma^\star a \right) G^2 \right] \\
&\overset{\text{(iii)}}{=} 2 \mathbb{E} \left[ (y + \bar{\eta}) \left( \bar{\gamma}^2 - \widetilde{\gamma}^2 \right) G^2 \right] \\
&\overset{\text{(iv)}}{\leq} 2 (y + \eta^\uparrow) \mathbb{E} \left[ \left( \bar{\gamma}^2 - \widetilde{\gamma}^2 \right) G^2 \mathbf{1} \{ \gamma^2 \geq \tau \} \right]
\end{aligned}
$$

where steps (i) and (ii) are true because $\bar{\gamma}^2 \geq \widetilde{\gamma}^2$ from (TP2), step (iii) follows from the definition $\bar{\gamma}^2 = \frac{y + \bar{\eta}}{a^\top \Sigma^\star a}$, and step (iv) follows from (TP3) and the definition of $\eta^\uparrow$ as the upper bound on the noise $\eta$.

The rest of the proof follows the exact steps of the proof of Lemma 7 in Appendix E.1.2. Therefore, we have the bound

$$\left\| \mathbb{E} \left[ \left( \bar{\gamma}^4 - \widetilde{\gamma}^4 \right) \left( a^\top \Sigma^\star a \right) a a^\top \right] \right\|_{\mathrm{op}} \lesssim \frac{y^\uparrow M^{1/2}}{\tau},$$

as desired.

### E.1.4 Proof of Lemma 9

The proof follows the steps as in the proof of Lemma 6, and we describe the difference of the two proofs. We again apply Bernstein's inequality.

**Proving a Bernstein condition.** We prove a Bernstein condition with $u_1 = c_1 (y + \eta^\uparrow)^2$ and $u_2 = c_2 (y + \eta^\uparrow) \tau$. Namely, for every integer $p \geq 2$, we have (cf. (31) in Lemma 6)

$$\mathbb{E} \left[ \left| v^\top \langle \widetilde{A}, \Sigma^\star \rangle \widetilde{A} v \right|^p \right] \leq \frac{p!}{2} u_1 u_2^{p-2}. \tag{37}$$

To show (37), we plug in $\widetilde{A} = \widetilde{\gamma}^2 a a^\top$ and have

$$
\begin{aligned}
\mathbb{E} \left| v^\top \langle \widetilde{A}, \Sigma^\star \rangle \widetilde{A} v \right|^p &= \mathbb{E} \left( \widetilde{\gamma}^2 a^\top \Sigma^\star a \right)^p \cdot \left| v^\top \widetilde{A} v \right|^p \\
&\overset{\text{(i)}}{\leq} \mathbb{E} \left( \bar{\gamma}^2 a^\top \Sigma^\star a \right)^p \cdot \left| v^\top \widetilde{A} v \right|^p \\
&\overset{\text{(ii)}}{=} \mathbb{E} \left( y + \bar{\eta} \right)^p \cdot \left| v^\top \widetilde{A} v \right|^p \\
&\overset{\text{(iii)}}{\leq} (y + \eta^\uparrow)^p \cdot \mathbb{E} \left| v^\top \widetilde{A} v \right|^p, \tag{38}
\end{aligned}
$$

where step (i) follows from (TP2), step (ii) follows from the definition $\bar{\gamma}^2 = \frac{y + \bar{\eta}}{a^\top \Sigma^\star a}$, and step (iii) follows from the definition of $\eta^\uparrow$ as the upper bound on the noise $\eta$. Substituting in (31) from Lemma 6 to bound the term $\mathbb{E} \left| v^\top \widetilde{A} v \right|^p$ in (38) completes the proof of the Bernstein condition (37).

**Applying Bernstein's inequality.** The rest of the proof follows in the same manner as the proof of Lemma 6 in Appendix E.1.1, with an additional factor of $(y + \eta^\uparrow)$. We have

$$\left\| \mathbb{E}\left[ \langle \widetilde{\boldsymbol{A}}, \boldsymbol{\Sigma}^\star \rangle \widetilde{\boldsymbol{A}} \right] - \frac{1}{n} \sum_{i=1}^n \langle \widetilde{\boldsymbol{A}}_i, \boldsymbol{\Sigma}^\star \rangle \widetilde{\boldsymbol{A}}_i \right\|_{\mathrm{op}} \lesssim y^\uparrow \left( \sqrt{\frac{M^{1/2}t}{n}} + \frac{\tau t}{n} \right)$$

with probability at least $1 - 2 \cdot 9^d \cdot \exp(-t)$, as desired.

### E.1.5   Proof of Lemma 10

Recall that by definition $\bar{\boldsymbol{A}} = \bar{\gamma}^2 \boldsymbol{a}\boldsymbol{a}^\top = \frac{y+\bar\eta}{\boldsymbol{a}^\top \boldsymbol{\Sigma}^\star \boldsymbol{a}} \boldsymbol{a}\boldsymbol{a}^\top$. We have

$$\left\| \mathbb{E}\left[ \bar\eta \bar{\boldsymbol{A}} \right] \right\|_{\mathrm{op}} = \left\| \mathbb{E}\left[ \bar\eta (y + \bar\eta) \frac{\boldsymbol{a}\boldsymbol{a}^\top}{\boldsymbol{a}^\top \boldsymbol{\Sigma}^\star \boldsymbol{a}} \right] \right\|_{\mathrm{op}}$$

$$= \left\| \mathbb{E}\left[ \bar\eta (y + \bar\eta) \right] \cdot \mathbb{E}\left[ \frac{\boldsymbol{a}\boldsymbol{a}^\top}{\boldsymbol{a}^\top \boldsymbol{\Sigma}^\star \boldsymbol{a}} \right] \right\|_{\mathrm{op}}$$

$$= \frac{\sigma_\eta^2}{m} \cdot \left\| \mathbb{E}\left[ \frac{\boldsymbol{a}\boldsymbol{a}^\top}{\boldsymbol{a}^\top \boldsymbol{\Sigma}^\star \boldsymbol{a}} \right] \right\|_{\mathrm{op}}. \tag{39}$$

To bound the operator norm term in (39), we apply Lemma 4(b) in Appendix D.4. For any matrix $\boldsymbol{U}$, we have

$$\mathbb{E}\left[ \frac{\boldsymbol{a}^\top \boldsymbol{U} \boldsymbol{a}}{\boldsymbol{a}^\top \boldsymbol{\Sigma}^\star \boldsymbol{a}} \right] \lesssim \frac{1}{\sigma_r r} \|\boldsymbol{U}\|_*. \tag{40}$$

Note that $\frac{\boldsymbol{a}\boldsymbol{a}^\top}{\boldsymbol{a}^\top \boldsymbol{\Sigma}^\star \boldsymbol{a}}$ is symmetric positive semidefinite, so we have

$$\left\| \mathbb{E}\left[ \frac{\boldsymbol{a}\boldsymbol{a}^\top}{\boldsymbol{a}^\top \boldsymbol{\Sigma}^\star \boldsymbol{a}} \right] \right\|_{\mathrm{op}} = \sup_{\boldsymbol{v} \in \mathcal{S}^{d-1}} \left| \boldsymbol{v}^\top \mathbb{E}\left[ \frac{\boldsymbol{a}\boldsymbol{a}^\top}{\boldsymbol{a}^\top \boldsymbol{\Sigma}^\star \boldsymbol{a}} \right] \boldsymbol{v} \right|$$

$$= \sup_{\boldsymbol{v} \in \mathcal{S}^{d-1}} \mathbb{E}\left[ \frac{\boldsymbol{a}^\top (\boldsymbol{v}\boldsymbol{v}^\top) \boldsymbol{a}}{\boldsymbol{a}^\top \boldsymbol{\Sigma}^\star \boldsymbol{a}} \right]$$

$$\overset{\text{(i)}}{\lesssim} \frac{1}{\sigma_r r} \sup_{\boldsymbol{v} \in \mathcal{S}^{d-1}} \|\boldsymbol{v}\boldsymbol{v}^\top\|_*$$

$$\overset{\text{(ii)}}{=} \frac{1}{\sigma_r r}, \tag{41}$$

where step (i) is true by substituting in (40) with $\boldsymbol{U} = \boldsymbol{v}\boldsymbol{v}^T$, and step (ii) is true because $\boldsymbol{v}$ is unit norm, and hence $\|\boldsymbol{v}\boldsymbol{v}^\top\|_* = 1$. Substituting (41) back to (39), we have

$$\left\| \mathbb{E}\left[ \bar\eta \bar{\boldsymbol{A}} \right] \right\|_{\mathrm{op}} \lesssim \frac{1}{\sigma_r r} \cdot \frac{\nu_\eta^2}{m},$$

as desired.

### E.2   Proof of Proposition 3

We analyze the term $\frac{1}{n} \sum_{i=1}^n \langle \widetilde{\boldsymbol{A}}_i, \boldsymbol{U} \rangle^2$ from (26). Recall from the definition of $\widetilde{\boldsymbol{A}}$ that for any $i = 1, \dots, n$,

$$\widetilde{\boldsymbol{A}}_i = \widetilde{\gamma}_i^2 \boldsymbol{a}_i \boldsymbol{a}_i^\top = \left( \frac{y + \bar\eta_i}{\boldsymbol{a}_i^\top \boldsymbol{\Sigma}^\star \boldsymbol{a}_i} \wedge \tau \right) \boldsymbol{a}_i \boldsymbol{a}_i^\top,$$

so we have

$$\langle \widetilde{\boldsymbol{A}}_i, \boldsymbol{U} \rangle^2 = \left( \frac{y + \bar\eta_i}{\boldsymbol{a}_i^\top \boldsymbol{\Sigma}^\star \boldsymbol{a}_i} \wedge \tau \right)^2 \left( \boldsymbol{a}_i^\top \boldsymbol{U} \boldsymbol{a}_i \right)^2. \tag{42}$$

From (42), we have that for any matrix $\boldsymbol{U}$, the term $\sum_{i=1}^{n} \langle \widetilde{\boldsymbol{A}}_i, \boldsymbol{U} \rangle^2$ is nondecreasing in $\tau$ when $\tau > 0$. Defining a random matrix

$$\widetilde{\boldsymbol{A}}^{\tau'} := \left( \frac{y + \bar{\eta}}{\boldsymbol{a}^\top \boldsymbol{\Sigma}^\star \boldsymbol{a}} \wedge \tau' \right) \boldsymbol{a} \boldsymbol{a}^\top, \tag{43}$$

for any $\tau' \in (0, \tau]$, we have

$$\frac{1}{n} \sum_{i=1}^{n} \langle \widetilde{\boldsymbol{A}}_i, \boldsymbol{U} \rangle^2 \geq \frac{1}{n} \sum_{i=1}^{n} \langle \widetilde{\boldsymbol{A}}_i^{\tau'}, \boldsymbol{U} \rangle^2, \tag{44}$$

where for every $i = 1, \ldots, n$, matrix $\widetilde{\boldsymbol{A}}_i^{\tau'}$ is formed with the same realizations of random quantities $\boldsymbol{a}_i$ and $\bar{\eta}_i$ as $\widetilde{\boldsymbol{A}}_i$. The two matrices only differ in choice of truncation threshold: $\tau'$ instead of $\tau$. As a result, for the rest of the proof, we lower bound $\frac{1}{n} \sum_{i=1}^{n} \langle \widetilde{\boldsymbol{A}}_i^{\tau'}, \boldsymbol{U} \rangle^2$ for an appropriate choice of $\tau'$ to be specified later. To proceed, we use a small-ball argument [49, 50] based on the following lemma.

**Lemma 11** ([50, Proposition 5.1], adapted to our notation). *Let $\boldsymbol{X}_1, \ldots, \boldsymbol{X}_n \in \mathbb{R}^{d \times d}$ be i.i.d. copies of a random matrix $\boldsymbol{X} \in \mathbb{R}^{d \times d}$. Let $E \subset \mathbb{R}^{d \times d}$ be a subset of matrices. Let $\xi > 0$ and $Q > 0$ be real values such that for every matrix $\boldsymbol{U} \in E$, the marginal tail condition holds:*

$$\mathbb{P}\left(|\langle \boldsymbol{X}, \boldsymbol{U} \rangle| \geq 2\xi\right) \geq Q. \tag{45}$$

*Define the Rademacher width as*

$$W := \mathbb{E}\left[ \sup_{\boldsymbol{U} \in E} \frac{1}{n} \sum_{i=1}^{n} \varepsilon_i \langle \boldsymbol{X}_i, \boldsymbol{U} \rangle \right],$$

*where $\varepsilon_1, \ldots, \varepsilon_n$ are i.i.d. Rademacher random variables independent of $\{\boldsymbol{X}_i\}_{i \in [n]}$. Then for any $t > 0$, we have*

$$\inf_{\boldsymbol{U} \in E} \left( \frac{1}{n} \sum_{i=1}^{n} \langle \boldsymbol{X}_i, \boldsymbol{U} \rangle^2 \right)^{1/2} \geq \xi(Q - t) - 2W.$$

*with probability at least $1 - \exp\left(-\frac{nt^2}{2}\right)$.*

Recall the error set $\mathcal{E}$ defined in Equation (22). Because the claim (26) is invariant to scaling, it suffices to prove it for $\|\boldsymbol{U}\|_F = 1$. Correspondingly, we define the set $E$ as

$$\begin{aligned} E &= \mathcal{E} \cap \{\boldsymbol{U} \in \mathbb{R}^{d \times d} : \|\boldsymbol{U}\|_F = 1\} \\ &= \{\boldsymbol{U} \in \mathbb{S}^{d \times d} : \|\boldsymbol{U}\|_F = 1, \|\boldsymbol{U}\|_* \leq 4\sqrt{2r}\}. \end{aligned} \tag{46}$$

We invoke Lemma 11 with set $E$ defined above, $\boldsymbol{X}_i = \widetilde{\boldsymbol{A}}_i^{\tau'}$, $\xi = \frac{c_1}{2} \left( \frac{\mu_y}{\operatorname{tr}(\boldsymbol{\Sigma}^\star)} \wedge \tau' \right)$, and $Q = c_2$, where $\mu_y$ is the median of $\bar{\eta}$ and $c_1$ and $c_2$, are constants to be specified later. The rest of the proof is comprised of two steps. We first verify that our choices for $\xi$ and $Q$ are valid for establishing the marginal tail condition (45). We then bound the Rademacher width $W$ above. The following lemma verifies our choices for $\xi$ and $Q$.

**Lemma 12.** *Consider any $\tau' \in (0, \tau]$. There exist absolute constants $c_1, c_2 > 0$ such that for every $\boldsymbol{U} \in E$, we have*

$$\mathbb{P}\left( \left| \langle \widetilde{\boldsymbol{A}}^{\tau'}, \boldsymbol{U} \rangle \right| \geq c_1 \left( \frac{\mu_y}{\operatorname{tr}(\boldsymbol{\Sigma}^\star)} \wedge \tau' \right) \right) \geq c_2.$$

The proof of Lemma 12 is presented in Appendix E.2.1. We now turn to the second step of the proof, which is bounding the Rademacher width $W$. The next lemma characterizes this width.

**Lemma 13.** *Consider any $\tau' \in (0, \tau]$. Let $\widetilde{\boldsymbol{A}}_1^{\tau'}, \ldots, \widetilde{\boldsymbol{A}}_n^{\tau'} \in \mathbb{R}^{d \times d}$ be i.i.d. copies of the random matrix $\widetilde{\boldsymbol{A}}^{\tau'} \in \mathbb{R}^{d \times d}$ defined in Equation (43). Let $E$ be the set defined in Equation (46). Then, there exists some absolute constants $c_1$ and $c_2$ such that if $n \geq c_1 d$, then we have*

$$\mathbb{E}\left[ \sup_{\boldsymbol{U} \in E} \frac{1}{n} \sum_{i=1}^{n} \varepsilon_i \langle \widetilde{\boldsymbol{A}}_i^{\tau'}, \boldsymbol{U} \rangle \right] \leq c_2 \tau' \sqrt{\frac{rd}{n}}.$$

The proof of Lemma 13 is presented in Appendix E.2.2. Lemma 12 establishes the marginal tail condition for Lemma 11, and Lemma 13 upper bounds the Rademacher width. We now invoke Lemma 11 and substitute in the upper bound for the Rademacher width $W$. For some constant $c_4$, if $n \geq c_4 d$, we have that with probability at least $1 - \exp\left(-\frac{nt^2}{2}\right)$,

$$
\inf_{U \in E} \left(\frac{1}{n} \sum_{i=1}^{n} \langle \widetilde{A}_i, U \rangle^2\right)^{1/2} \overset{\text{(i)}}{\geq} \inf_{U \in E} \left(\frac{1}{n} \sum_{i=1}^{n} \langle \widetilde{A}_i^{\tau'}, U \rangle^2\right)^{1/2}
$$

$$
\geq \frac{c_1}{2} \left(\frac{\mu_y}{\text{tr}\left(\Sigma^\star\right)} \wedge \tau'\right) (c_2 - t) - c_3 \tau' \sqrt{\frac{rd}{n}},
$$

where step (i) is true due to the monotonicity property (44). We set $\tau' = \frac{\mu_y}{\text{tr}(\Sigma^\star)}$, where recall that $\mu_y$ is the median of the random quantity $y + \bar{\eta}$. By the assumption $\tau \geq \frac{\mu_y}{\text{tr}(\Sigma^\star)}$, this choice of $\tau'$ satisfies $\tau' \leq \tau$. Setting $t = \frac{c_2}{2}$, we have that with probability at least $1 - \exp\left(-\frac{c_2^2 n}{8}\right)$,

$$
\inf_{U \in E} \frac{1}{n} \left(\sum_{i=1}^{n} \langle \widetilde{A}_i^{\tau'}, U \rangle^2\right)^{1/2} \geq \frac{c_1 c_2}{4} \frac{\mu_y}{\text{tr}\left(\Sigma^\star\right)} - c_3 \frac{\mu_y}{\text{tr}\left(\Sigma^\star\right)} \sqrt{\frac{rd}{n}}.
$$

Recall from the definition of $E$ (46) that $\|U\|_F = 1$. As a result, if $n \geq \max\left\{\left(\frac{4c_3}{c_1 c_2}\right)^2, c_4\right\} rd$, we have

$$
\inf_{U \in \mathcal{E}} \frac{1}{n} \sum_{i=1}^{n} \langle \widetilde{A}_i, U \rangle^2 \geq \left(\frac{c_1 c_2}{4} \frac{\mu_y}{\text{tr}\left(\Sigma^\star\right)}\right)^2 \|U\|_F^2
$$

with probability at least $1 - \exp\left(-\frac{c_2^2 n}{8}\right)$. We conclude by setting $\kappa_{\mathcal{L}} = \left(\frac{c_1 c_2}{4}\right)^2$, $c = \frac{c_2^2}{8}$, and $C = \max\left\{\left(\frac{4c_3}{c_1 c_2}\right)^2, c_4\right\}$ in Proposition 3.

### E.2.1   Proof of Lemma 12

We fix any $U \in E$. Recall that $\mu_y$ denotes the median of $y + \bar{\eta}$. Let $\mathcal{G}$ be the event that $y + \bar{\eta} \geq \mu_y$, which occurs with probability $\frac{1}{2}$. For any $\xi > 0$, because the averaged noise $\bar{\eta}$ and sensing vector $a$ are independent, we have

$$
\mathbb{P}\left(\left|\langle \widetilde{A}^{\tau'}, U \rangle\right| \geq \xi\right) \overset{\text{(i)}}{=} \mathbb{P}\left(\left(\frac{y + \bar{\eta}}{a^\top \Sigma^\star a} \wedge \tau'\right) \cdot \left|\langle aa^\top, U \rangle\right| \geq \xi\right)
$$

$$
= \mathbb{P}\left(\left(\frac{y + \bar{\eta}}{a^\top \Sigma^\star a} \wedge \tau'\right) \cdot \left|\langle aa^\top, U \rangle\right| \geq \xi \,\middle|\, \mathcal{G}\right) \cdot \mathbb{P}\left(\mathcal{G}\right)
$$

$$
= \frac{1}{2} \mathbb{P}\left(\left(\frac{y + \bar{\eta}}{a^\top \Sigma^\star a} \wedge \tau'\right) \cdot \left|\langle aa^\top, U \rangle\right| \geq \xi \,\middle|\, \mathcal{G}\right)
$$

$$
\overset{\text{(ii)}}{\geq} \frac{1}{2} \mathbb{P}\left(\left(\frac{\mu_y}{a^\top \Sigma^\star a} \wedge \tau'\right) \cdot \left|\langle aa^\top, U \rangle\right| \geq \xi\right), \tag{47}
$$

where step (i) is true by plugging in the definition of $\widetilde{A}^{\tau'}$, and step (ii) is true by the definition of the event $\mathcal{G}$. We proceed by bounding the terms in (47) separately.

**Lower bound on $\left|\langle aa^\top, U \rangle\right|$.**   We use the approach from [32, Section 4.1]. By Paley-Zygmund inequality,

$$
\mathbb{P}\left(\left|\langle aa^\top, U \rangle\right|^2 \geq \frac{1}{2} \mathbb{E}\left[\left|\langle aa^\top, U \rangle\right|^2\right]\right) \geq \frac{1}{4} \frac{\left(\mathbb{E}\left[\left|\langle aa^\top, U \rangle\right|^2\right]\right)^2}{\mathbb{E}\left[\left|\langle aa^\top, U \rangle\right|^4\right]} \tag{48}
$$

We now analyze the terms in (48). As noted in [32, Section 4.1], there exists some constant $c_1 > 0$ such that for any matrix $U$ with $\|U\|_F = 1$,

$$\mathbb{E}\left[|\langle aa^\top, U\rangle|^2\right] \geq 1 \quad \text{and} \quad \mathbb{E}\left[|\langle aa^\top, U\rangle|^4\right] \leq c_1 \left(\mathbb{E}\left[|\langle aa^\top, U\rangle|^2\right]\right)^2. \tag{49}$$

Note that by the definition of the set $E$, every matrix $U \in E$ satisfies $\|U\|_F = 1$. Utilizing inequalities (48) and (49), there exists positive constant $c_2 > 0$ such that

$$\mathbb{P}\left(|\langle aa^\top, U\rangle| \geq \frac{1}{2}\right) \geq c_2. \tag{50}$$

**Upper bound on $a^\top \Sigma^\star a$.** By Hanson-Wright inequality [51, Theorem 1.1], there exist some positive absolute constants $c_3$ and $c_4$ such that for any $t > 0$, we have

$$\mathbb{P}\left(a^\top \Sigma^\star a \leq c_3\left(\text{tr}\left(\Sigma^\star\right) + \|\Sigma^\star\|_F \sqrt{t} + \|\Sigma^\star\|_{\text{op}} t\right)\right) \geq 1 - 2\exp\left(-c_4 t\right).$$

We set $t = -\frac{1}{c_4}\log(\frac{c_2}{4})$ so that $2\exp\left(-c_4 t\right) = \frac{c_2}{2}$. Since $\Sigma^\star$ is symmetric positive semidefinite, we have

$$\|\Sigma^\star\|_F \leq \text{tr}\left(\Sigma^\star\right)$$
$$\text{and } \|\Sigma^\star\|_{\text{op}} \leq \text{tr}\left(\Sigma^\star\right)$$

As a result, we have that there exists some constant $c_5 > 0$ such that

$$\mathbb{P}\left(a^\top \Sigma^\star a \leq c_5 \text{tr}\left(\Sigma^\star\right)\right) \geq 1 - \frac{c_2}{2}. \tag{51}$$

**Substituting the two bounds back to** (47). By a union bound of (50) and (51), we have

$$\mathbb{P}\left(\left(\frac{\mu_y}{a^\top \Sigma^\star a} \wedge \tau'\right) \cdot |\langle aa^\top, U\rangle| \geq \frac{1}{2}\left(\frac{\mu_y}{c_5 \text{tr}\left(\Sigma^\star\right)} \wedge \tau'\right)\right)$$

$$\geq \mathbb{P}\left(\frac{\mu_y}{a^\top \Sigma^\star a} \wedge \tau' \geq \frac{\mu_y}{c_5 \text{tr}\left(\Sigma^\star\right)} \wedge \tau'\right) + \mathbb{P}\left(|\langle aa^\top, U\rangle| \geq \frac{1}{2}\right) - 1$$

$$\geq \mathbb{P}\left(\frac{\mu_y}{a^\top \Sigma^\star a} \geq \frac{\mu_y}{c_5 \text{tr}\left(\Sigma^\star\right)}\right) + \mathbb{P}\left(|\langle aa^\top, U\rangle| \geq \frac{1}{2}\right) - 1 \geq \frac{c_2}{2} \tag{52}$$

Combining (52) and (47), and redefining constant $c_2$ appropriately, we have

$$\mathbb{P}\left(\left|\langle \widetilde{A}^{\tau'}, U\rangle\right| \geq \frac{1}{2}\left(\frac{\mu_y}{\text{tr}\left(\Sigma^\star\right)} \wedge \tau'\right)\right) \geq c_2,$$

as desired.

### E.2.2  Proof of Lemma 13

We begin by noting that for any matrix $U \in E$,

$$\mathbb{E}\left[\sup_{U \in E} \frac{1}{n}\sum_{i=1}^n \varepsilon_i \langle \widetilde{A}_i^{\tau'}, U\rangle\right] \overset{(i)}{\leq} \mathbb{E}\left[\sup_{U \in E} \left\|\frac{1}{n}\sum_{i=1}^n \varepsilon_i \widetilde{A}_i^{\tau'}\right\|_{\text{op}} \cdot \|U\|_*\right]$$

$$\overset{(ii)}{\leq} 4\sqrt{2r} \cdot \mathbb{E}\left\|\frac{1}{n}\sum_{i=1}^n \varepsilon_i \widetilde{A}_i^{\tau'}\right\|_{\text{op}}, \tag{53}$$

where step (i) follows from Hölder's inequality, and step (ii) follows from the fact that $\|U\|_* \leq 4\sqrt{2r}$ from the definition of the set $E$. It remains to bound the expectation of the operator norm in (53). We follow the standard covering arguments in [52, Section 5.4.1], [50, Section 8.6], [32, Section 4.1], with a slight modification to accommodate the bounded term $\left(\frac{y+\bar{\eta}_i}{a_i^\top \Sigma^\star a_i} \wedge \tau'\right)$ that appears in each of the matrices $\widetilde{A}_i^{\tau'}$. As a result, there exist universal constants $c_1$ and $c_2$ such that if $n$ satisfies $n \geq c_1 d$, then we have

$$\mathbb{E}\left[\left\|\frac{1}{n}\sum_{i=1}^n \varepsilon_i \widetilde{A}_i^{\tau'}\right\|_{\text{op}}\right] \leq c_2 \tau' \sqrt{\frac{d}{n}}.$$

We conclude by re-defining $c_2$ appropriately.

# F    Proof of Corollary 1

We proceed by considering two cases. For each case, the proof consists of two steps. We first verify that the choices of the averaging parameter $m$ and truncation threshold $\tau$,

$$m = \left\lceil \left[ \left( \frac{\nu_\eta^2}{(y^\uparrow)^2} \right)^2 \frac{N}{d} \right]^{1/3} \right\rceil \quad \text{and} \quad \tau = \frac{y^\uparrow}{\sigma_r r} \sqrt{\frac{N}{md}}, \tag{54}$$

satisfy the assumptions of Theorem 1, namely $n \geq C_2 rd$ and $\tau \geq \frac{\mu_y}{\text{tr}(\boldsymbol{\Sigma}^\star)}$. We then invoke Theorem 1.

## F.1    Case 1: high-noise regime

In this case, we have $\frac{\nu_\eta^2}{(y^\uparrow)^2} > \sqrt{\frac{d}{N}}$, which means by setting $m$ according to Equation (54), we have $m \geq 2$. As a result, the bound

$$\left\lceil \left[ \left( \frac{\nu_\eta^2}{(y^\uparrow)^2} \right)^2 \frac{N}{d} \right]^{1/3} \right\rceil \leq 2 \left[ \left( \frac{\nu_\eta^2}{(y^\uparrow)^2} \right)^2 \frac{N}{d} \right]^{1/3} \tag{55}$$

holds in the high-noise regime.

**Verifying the condition on $n$.**    Recall that $n = \frac{N}{m}$. We have

$$n = \frac{N}{m} \overset{(i)}{\geq} \frac{N}{2} \left( \left( \frac{\nu_\eta^2}{(y^\uparrow)^2} \right)^2 \frac{N}{d} \right)^{-1/3}$$

$$= \frac{1}{2} \left( N^2 d \left( \frac{(y^\uparrow)^2}{\nu_\eta^2} \right)^2 \right)^{1/3}$$

$$\overset{(ii)}{\geq} \left( C_2^3 \left( \frac{(y^\uparrow)^2}{\nu_\eta^2} \right)^2 \left( \frac{\nu_\eta^2}{(y^\uparrow)^2} \right)^2 r^3 d^3 \right)^{1/3}$$

$$= C_2 rd,$$

where step (i) is true by plugging in the choice of $m$ from (54) and applying the bound (55), and step (ii) is true by substituting in the assumption $N \geq 2C_2^{3/2} \left( \frac{\nu_\eta^2}{(y^\uparrow)^2} \right)^2 r^{3/2} d$. Thus the condition $n \geq C_2 rd$ of Theorem 1 is satisfied.

**Verifying the condition on $\tau$.**    For the term $\sqrt{\frac{N}{dm}}$ in the expression of $\tau$ in (54), note that, by the previous point, $\frac{N}{m} = n \gtrsim rd$ (with a constant that is greater than 1). Thus $\sqrt{\frac{N}{dm}} \geq \sqrt{r} > 1$. Therefore, to verify the condition $\tau \geq \frac{\mu_y}{\text{tr}(\boldsymbol{\Sigma}^\star)}$, it suffices to verify that

$$\frac{y^\uparrow}{\sigma_r r} \geq \frac{\mu_y}{\text{tr}(\boldsymbol{\Sigma}^\star)}. \tag{56}$$

By definition, we have $y^\uparrow \geq \mu_y$. Furthermore, since $\boldsymbol{\Sigma}^\star$ is symmetric positive semidefinite, its eigenvalues are all non-negative and are identical to its singular values, and hence $\text{tr}(\boldsymbol{\Sigma}^\star) \geq \sigma_r r$, verifying the condition (56).

**Invoking Theorem 1.**    By setting $\lambda_n$ to its lower bound in (9) and substituting in $n = \frac{N}{m}$ and our choice of $\tau$ from (54), we have

$$\lambda_n = C_1 \left( 3 \frac{(y^\uparrow)^2}{\sigma_r r} \sqrt{\frac{md}{N}} + \frac{\nu_\eta^2}{m} \right) \tag{57}$$

Substituting this expression of $\lambda_n$ to the error bound (10), then substituting in our choice of $m$ from (54) to (57) and defining $C' = 3C \cdot C_1$, we have

$$\|\widehat{\boldsymbol{\Sigma}} - \boldsymbol{\Sigma}^\star\|_F \leq C' \left( \frac{\mathrm{tr}\,(\boldsymbol{\Sigma}^\star)^2}{\sigma_r r} \right) \frac{(y^\uparrow)^{4/3}(\nu_\eta^2)^{1/3}}{\mu_y^2} \sqrt{r} \left( \frac{d}{N} \right)^{1/3}.$$

Using the fact that $\mathrm{tr}\,(\boldsymbol{\Sigma}^\star) \leq \sigma_1 r$, we have

$$\|\widehat{\boldsymbol{\Sigma}} - \boldsymbol{\Sigma}^\star\|_F \leq C' \frac{\sigma_1^2}{\sigma_r} \frac{(y^\uparrow)^{4/3}(\nu_\eta^2)^{1/3}}{\mu_y^2} r^{3/2} \left( \frac{d}{N} \right)^{1/3}$$

as desired.

### F.2  Case 2: low-noise regime

In this case, we have $\frac{\nu_\eta^2}{(y^\uparrow)^2} \leq \sqrt{\frac{d}{N}}$, which means by setting $m$ according to Equation (54), we have $m = 1$. As a result, no averaging occurs.

**Verifying the condition on $n$.**  Because $m = 1$ in this case, we have that $n = N$. By assumption, we have that $N \geq C_2 rd$, satisfying the condition $n \geq C_2 rd$ in Theorem 1.

**Verifying the condition on $\tau$.**  By the same analysis as in Case 1, we have that the condition $\tau \geq \frac{\mu_y}{tr \boldsymbol{\Sigma}^\star}$ in Theorem 1.

**Invoking Theorem 1.**  By setting $\lambda_n$ to its lower bound in (9), substituting in our choice of $\tau$ from (54) and noting $m = 1$, we have

$$\lambda_n = C_1 \left( 3 \frac{(y^\uparrow)^2}{\sigma_r r} \sqrt{\frac{d}{n}} + \frac{1}{\sigma_r r} \nu_\eta^2 \right). \tag{58}$$

We define $C' = 3C \cdot C_1$ and note that $n = N$ under Case 2. Substituting this expression of $\lambda_n$ in (58) to the error bound (10), then using the fact that under Case 2, the bound $\nu_\eta^2 \leq (y^\uparrow)^2 \sqrt{\frac{d}{N}}$ holds, we have

$$\|\widehat{\boldsymbol{\Sigma}} - \boldsymbol{\Sigma}^\star\|_F \leq C' \left( \frac{\mathrm{tr}\,(\boldsymbol{\Sigma}^\star)^2}{\sigma_r r} \right) \left( \frac{y^\uparrow}{\mu_y} \right)^2 \sqrt{\frac{rd}{N}}.$$

Using the fact that $\mathrm{tr}\,(\boldsymbol{\Sigma}^\star) \leq \sigma_1 r$, we have

$$\|\widehat{\boldsymbol{\Sigma}} - \boldsymbol{\Sigma}^\star\|_F \leq C' \frac{\sigma_1^2}{\sigma_r} \left( \frac{y^\uparrow}{\mu_y} \right)^2 \sqrt{\frac{r^3 d}{N}},$$

as desired.

