# OpenReview forum: "Perceptual adjustment queries and an inverted measurement paradigm for low-rank metric learning"
_NeurIPS.cc/2023/Conference — NeurIPS 2023 poster_

### Official Review · Reviewer_H5Pr · 2023-06-17

**Soundness:** 2 fair
**Presentation:** 3 good
**Contribution:** 2 fair
**Rating:** 5
**Confidence:** 3

**Summary:**

This paper introduces perceptual adjustment query (PAQ), which combines cardinal reporting and ordinal reasoning using a sliding bar for relational questions through identifying the transition point (similar vs. dissimilar) along a continuum to the reference item. Formulating the problem as Mahalanobis distance learning in the feature space, this gives rise to a high-dimensional, low-rank matrix estimation problem. The authors propose two techniques including sample averaging and truncation to overcome challenges not present in classical low-rank matrix estimation problems and prove statistical bounds on the estimator error.

**Strengths:**

1. This paper studies the interesting problem of collecting more expressive and self-consistent human responses without heavy cognitive burden on users. The proposed PAQ method appears to achieve this by combining the advantages of both cardinal and ordinal queries. This gives a continuous way of constructing the tuplewise similarity queries [1]. And instead of asking for similarity, it asks for the transition point from similar to dissimilar.

2. Thanks to the new approach for collecting human responses, the formulated Mahalanobis metric learning problem encounters several new challenges, including dependent noise and a heavy-tailed sensing matrix. To address these issues, the authors propose to leverage the averaging and truncation techniques. Both theoretical and simulation analyses demonstrate the efficacy of these methods.

3. This paper presents a coherent structure and is easy to follow.

[1] G. Canal, S. Fenu, and C. Rozell. Active ordinal querying for tuplewise similarity learning. AAAI 2020.

**Weaknesses:**

1. One major concern regarding this work is the lack of empirical validation for the proposed PAQ in real-world applications. The authors offer a conceptual explanation of its functionality and user interface design but fail to provide any implementation guidance. For instance, the practical implementation of the continuum remains unclear - should it be continuous or discretized? Moreover, the authors do not provide any synthetic tests or real human responses to support the claimed benefits or practical applicability of the proposed PAQ. The current validation relies solely on simulation results, which assess the theoretical analysis in relatively low-dimensional settings instead of truly high-dimensional scenarios.

2. Another concern is the dependence of the proposed method on a pre-trained generative model that exhibits a linear latent space. This constraint limits the method's broad applicability compared to existing methods, such as tuplewise queries [1]. Also, many items generated along the line may be invalid in specific domains, such as molecules and proteins.

3. The proposed solutions to the two challenges seem somewhat expected: averaging to reduce the variance and truncation to mitigate the heavy-tail issue have been commonly used in similar situations.

**Questions:**

1. What are the main contributions of this work? The authors claim both the proposal of the PAQ for the Mahalanobis metric learning and the solution to the associated matrix estimation problem in Section 1.1. Therefore, both should be validated quantitatively to make this work coherent and complete.

2. The definition of the perceptual adjustment query in Section 2.2 does not seem to be dependent on a latent space associated with a item decoder/generator but only a feature space. Could the PAQ be applied to items solely within a feature space where a decoder/generator is not feasible, such as in the case of housing pricing similarity?

3. The authors mention in Section 5 that they "cross-validate to chose the constant factor." How to perform cross-validation in practice, given that only human responses and query vectors are available? Also, the boundary value y is set at 200, which seems to be an unusually large number. Although the actual value of y may not be critical due to the scale invariance property, the noise variance scales with y.

4. The authors mention that they "develop a two-stage estimator for metric learning from PAQs". What are those two stages? From Algorithm 1, there seems to be only one stage.

**Limitations:**

The authors provide a discussion of limitations and potential societal impact in Appendix A.1 and A.2.

---

> ### Author Rebuttal · Authors · 2023-08-10
>
> Thank you for your comments and feedback. We are grateful that you appreciate the advantages of our proposed query and the writing of the paper.
>
> > What are the main contributions of this work?
>
> Please see the global response for a detailed response. In summary, our paper has three main contributions: The PAQ, the measurement scheme that arises when applying PAQs to low-rank metric learning, and the resulting estimator and corresponding theoretical guarantees. We emphasize the latter two contributions are of independent methodological interest to the low-rank matrix estimation community.
>
> > The current validation relies solely on simulation results, which assess the theoretical analysis in relatively low-dimensional settings instead of truly high-dimensional scenarios.
>
> >The authors claim both the proposal of the PAQ for the Mahalanobis metric learning and the solution to the associated matrix estimation problem in Section 1.1. Therefore, both should be validated quantitatively to make this work coherent and complete.
>
> We have performed additional experiments comparing PAQs against two often used queries in metric learning: Triplets and binary comparisons. For full results, please see the global response and the attached PDF. To summarize, PAQs drastically outperform the two baselines, greatly reducing the number of queries needed to achieve a particular error level.
>
> > The proposed solutions to the two challenges seem somewhat expected: averaging to reduce the variance and truncation to mitigate the heavy-tail issue have been commonly used in similar situations.
>
> While averaging and truncation has been used in other statistics problems, the employment of these techniques was not obvious for two reasons:
>
> 1.) The proposed solutions are in response to the simultaneous occurrence of two problems: estimator bias and heavy-tailed design. To our knowledge, no work has addressed both of these problems simultaneously in the low-rank sensing literature, meaning our contribution of the average-then-truncate estimator is completely novel.
>
>  2.) The averaging step is utilized to ensure our estimator is consistent, not simply to reduce variance. If the averaging step is omitted, the estimation error remains bounded away from zero no matter how large N grows, i.e., the estimator without the averaging step is biased.
>
> Considering practical points made estimator design even more challenging: We must design an estimator that does not impose any burdens at response time. To this point, both averaging and truncation steps are applied after collecting user responses, meaning no additional action is required from the user to ensure consistent estimation.
>
> > Another concern is the dependence of the proposed method on a pre-trained generative model that exhibits a linear latent space. This constraint limits the method's broad applicability...
>
> We emphasize that the latent space of a generative model is an example setting, but the PAQ is not constrained to this setting. We analyze the PAQ under the case where the user traverses the continuous feature space in a linear manner, and define the query for this analysis setting. However, in practice, there is no requirement that the path of points presented to a user be a straight line. We consider designing informative paths as an interesting avenue for future work.
>
> > Could the PAQ be applied to items solely within a feature space where a decoder/generator is not feasible, such as in the case of housing pricing similarity?
>
> Yes. We emphasize that the PAQ is broadly applicable, including the application of housing price similarity (see global response). The PAQ can be in a variety of settings, notably any continuous valued feature space.
>
> > One major concern regarding this work is the lack of empirical validation for the proposed PAQ in real-world applications. The authors offer a conceptual explanation of its functionality and user interface design but fail to provide any implementation guidance. For instance, the practical implementation of the continuum remains unclear - should it be continuous or discretized?
>
> In practice, these details are left to practitioners to adapt to their specific use cases. However, we will be sure to include a broad discussion of potential practical details in our revision along with a detailed discussion of example use cases.
>
> Towards the specific point of discrete vs. continuous implementation, we note that the implementation can be either. We focus on the continuous case in the paper for analysis purposes, but note that discretizing the path of points can be viewed as a form of bounded noise (with discrete support) in our measurements, with the noise being bounded by the quantization level. Since our theoretical results only assume bounded noise, they cover this discrete setting too.
>
> > The authors mention in Section 5 that they "cross-validate to chose the constant factor." How to perform cross-validation in practice, given that only human responses and query vectors are available?
>
> In practice, one may perform (cross-)validation as follows. First, we collect responses from a human and split it up into a train/validation set. Then we apply our estimator to the training set, employing averaging and truncation, and solving the regularized optimization problem. We can then choose our hyperparameters to minimize validation error. Note that in this cross-validation procedure, we only use human responses and query vectors, i.e., no “ground-truth” is needed.
>
> > The authors mention that they "develop a two-stage estimator for metric learning from PAQs". What are those two stages? From Algorithm 1, there seems to be only one stage.
>
> The two stages refer to 1.) preprocessing (i.e., averaging and truncation) and 2.) solving the regularized estimation problem. We will clarify this in the camera-ready version.

---

> > ### Comment · Reviewer_H5Pr · 2023-08-15
> >
> > Thanks for the response, which has addressed some of my concerns. However, there are still some unresolved. The added simulation comparison with triplet and binary comparisons is nice, but it is still much better to compare with a more similar tuplewise similarity queries [1] and perform human evaluations to demonstrate the broad applicability of the work in real-world settings similar to the experiment performed in [1]. In addition, the authors emphasized both in the paper and the rebuttal that they are working with "high-dimensional" data, but the largest number of dimensions tested is 60 which is hardly high-dimensional. In image embeddings, the number of dimensions can easily exceed 1000.

---

> > > ### Author Response · Authors · 2023-08-16
> > >
> > > We thank the reviewer for their response. Below, we address their remaining concerns.
> > >
> > > > it is still much better to compare with a more similar tuplewise similarity queries [1]
> > >
> > > We have performed an additional set of simulations comparing PAQs to the tuplewise ranking queries proposed in [1]. As the authors of [1] mention, to perform inference with these tuplewise ranking queries, one decomposes the ranking query response into a set of triplet responses. Thus, informativeness of ranking queries is relatively limited in comparison to PAQs, in that responses quantize statements of human perception. Furthermore, as the size of the set of items to be ranked grows, the cognitive burden on the user increases dramatically [2], [3]
> > >
> > > Plots of relative error vs. number of queries look similar to the already attached figures but we are not allowed to attach updated figures; instead, please see the table below for our results. We denote ranking-$k$ to mean $k$ items are ranked in relation to a reference object, meaning $k+1$ total items are presented to a user, and report normalized estimation error.
> > >
> > > | Number of responses | 500       | 1000      | 1500      | 2000      | 2500      |
> > > |---------------------|-----------|-----------|-----------|-----------|-----------|
> > > | PAQ                 | **0.396** | **0.135** | **0.094** | **0.073** | **0.063** |
> > > | ranking-$3$         | 0.621     | 0.398     | 0.309     | 0.253     | 0.225     |
> > > | ranking-$4$         | 0.423     | 0.268     | 0.210     | 0.178     | 0.156     |
> > >
> > > In summary, PAQs still outperform the more complex tuplewise ranking queries whenever a reasonably small error is desired, i.e., to achieve reasonable estimation error, one needs substantially more ranking query responses than PAQ responses. For example, to achieve normalized estimation error of 0.2 in the noisy regime, one needs
> > > - PAQ: Between 700 - 800 responses
> > > - ranking-$3$: Greater than 2500 responses
> > > - ranking-$4$: Between 1500 - 2000 responses
> > > - Triplets: Greater than 2500 responses
> > >
> > >  > perform human evaluations to demonstrate the broad applicability of the work in real-world settings similar to the experiment performed in [1].
> > >
> > > We emphasize that our contributions are orthogonal to those of [1]. In particular, while [1] develops a new query and performs real-world experiments with human responses, they do not develop any theoretical guarantees for estimation error with said query responses. On the other hand, a core part of our contribution are theoretical and mathematical developments: We (1) surface a previously unstudied measurement model for low-rank matrix estimation, (2) develop a novel estimator, and (3) prove theoretical bounds that apply in the high-dimensional setting. In doing so, we took care to set up the scope of the paper to highlight these contributions in Section 1.1. We agree that human evaluations will make a valuable future contribution, but they fall outside the scope of this work.
> > >
> > >
> > > > the authors emphasized both in the paper and the rebuttal that they are working with "high-dimensional" data, but the largest number of dimensions tested is 60 which is hardly high-dimensional.
> > >
> > > The runtime complexity of convex semidefinite programs scales cubically with dimension, making solving repeated trials computationally expensive. As a result, it is accepted practice in the low-rank matrix estimation literature to experiment with dimensions comparable to ours. For example, [4] uses $d = 40, 50, 60$, [5] uses $d = 40$, and [6] uses $d = 40, 80, 160$. We note that when deploying PAQs in practice in high-dimensions, one can utilize scalable non-convex optimization approaches that optimize low-rank matrix factors directly. These methods have been rigorously shown to converge in related settings (see [7], [8]).
> > >
> > > ---
> > >
> > > [1] G. Canal, S. Fenu, C. Rozell. ​​”Active Ordinal Querying for Tuplewise Similarity Learning,” in AAAI 2020
> > >
> > > [2] H. A. Simon, “A behavioral model of rational choice,” The quarterly journal of economics, vol. 69, no. 1, pp. 99–118, 1955.
> > >
> > > [3] G. Miller, “The magical number seven, plus or minus two: Some limits on our capacity for processing information.” Psych. Rev., vol. 63, no. 2, p. 81, 1956.
> > >
> > > [4]  J. Fan, W. Wang, Z. Zhu, “A shrinkage principle for heavy-tailed data: High-dimensional robust low-rank matrix recovery,” in the Annals of Statistics 2021.
> > >
> > > [5] Y. Chen, Y. Chi, A. Goldsmith, "Exact and stable covariance estimation from quadratic sampling via convex programming," in IEEE Transactions on Information Theory 2015
> > >
> > > [6] S. Negahban, M. Wainwright. “Estimation of (near) low-rank matrices with noise and high-dimensional scaling,” in the Annals of Statistics 2011
> > >
> > > [7] Y. Dong, M. Wainwright. “Fast low-rank estimation by projected gradient descent: General statistical and algorithmic guarantees,” arxiv
> > >
> > > [8] S. Tu, R. Boczar, M. Simchowitz, M. Soltanolkotabi, B. Recht, “Low-rank Solutions of Linear Matrix Equations via Procrustes Flow,” in ICML 2016

---

> > > > ### Comment · Reviewer_H5Pr · 2023-08-17
> > > >
> > > > Thanks for the additional quantitative results. I have raised my score.

---

### Official Review · Reviewer_eVNV · 2023-07-04

**Soundness:** 2 fair
**Presentation:** 2 fair
**Contribution:** 3 good
**Rating:** 4
**Confidence:** 4

**Summary:**

People evaluate things in two ways: scoring and comparing, which is the process of generating cardinal data and ordinal data. However, since different people have different scales, the cardinal data are more influenced by people's subjective perception, while the ordinal data are fundamentally limited and laces expressiveness because the comparison results are only for several things. Therefore, the paper introduces a new type of informative and yet cognitively lightweight evaluation mechanism for collecting human feedback. Specifically, by generating a series of consecutive items between two things, one progressively compares the reference item with other items, making cognitive judgments in a relative sense while identifying a transition point (similar vs. dissimilar) in the ranking.  Subsequently, the Mahalanobis distance metric learning model is proposed based on the above evaluation mechanism, where the model requires that the learned metric makes each reference sample as close as possible to its generated transition point. Moreover, the generalized error bound analysis of the proposed model is given.

**Strengths:**

The idea of the paper is novel. The paper presents a novel query mechanism and uses it to propose an efficient and easily solvable metric learning model.

**Weaknesses:**

The idea of the paper is novel, however the experiments are very inadequate and not at all compared with any known metric learning algorithms. It is suggested to add comparative experiments with existing metric learning algorithms under different perspectives to fully illustrate the value of the proposed algorithm. Meanwhile, there are some expression and formatting problems with the paper. The details are as follows:
1. Lack of description of existing relevant metric learning.
2. There are major problems with the format of references, such as the lack of journal names and page numbers in some citations, as well as the problem of inconsistent cases between the paper name and the journal name.
3 There are inconsistencies in expression or format. For example, “metric model” and “measurement model”, “In metric learning” and “In our theoretical results”.
4 The usage of some symbols is inconsistent with the conventions of metric learning. For example, y often indicates a label in metric learning, there is a symbol with an arrow to indicate a constant that has not seen such usage, and there is a greater than or equal symbol with a wavy line that does not understand its meaning.
5. There are some grammatical errors, please proofread carefully. For example, “such as in conducting surveys, grading assignments,…”should remove “in”, inconsistent case after the initial letter of the colon.

**Questions:**

1.  While novel ideas and theories are important, it is sufficient experimentation that will prove the value of the proposed method. It is recommended that sufficient experiments be added to the text to fully demonstrate the significance of the proposed model.
2.  Comparison with existing metric learning and discussion are lacking, and it is suggested to add relevant literature descriptions to highlight the contribution of the proposed method.

**Limitations:**

The idea is novel, the model is simple, and it is suggested that the proposed method be extended to more complex multi-metric and deep metric learning models to enhance the usefulness of the proposed theory and model.

---

> ### Author Rebuttal · Authors · 2023-08-10
>
> Thank you for your comments and feedback. We are grateful that you appreciate the novelty of our proposed query.
>
> >While novel ideas and theories are important, it is sufficient experimentation that will prove the value of the proposed method. It is recommended that sufficient experiments be added to the text to fully demonstrate the significance of the proposed model.
>
> >Comparison with existing metric learning … are lacking
>
>
> We emphasize that the focus of this work is to provide fundamental understanding to the proposed new query type, as opposed to empirical improvement on certain tasks or datasets. We chose to limit the scope of this work to metric learning precisely to highlight these unique theoretical challenges, and provide solutions that are of independent interest to the statistics community.
>
> Specifically, as mentioned in the global response, the focus on metric learning allows us to bring two novel problems to the high-dimensional statistics and low-rank matrix sensing communities: the un-studied inverted measurement scheme and estimator design under simultaneously heavy-tailed and bias-inducing sensing matrices. We present, to the best of our knowledge, the first estimator under these novel problem settings.
>
> To further our fundamental understanding, we have performed a new set of simulation experiments comparing PAQs against two often used queries in metric learning: Triplets and binary comparisons (please see the global response and the attached PDF for full results). We show that PAQs drastically outperform the two baselines, greatly reducing the number of queries needed to achieve a particular error level.
>
>
> > Lack of description of existing relevant metric learning.
>
> > It is suggested to add relevant literature descriptions to highlight the contribution of the proposed method.
>
> Metric learning has a rich history in machine learning and signal processing, and we limit ourselves to discussing the literature that is most closely related to our contribution. In hindsight, we see that the choice of title for this subsection (“Metric learning”) may be overly broad, and will change it in a future revision to read “Mahalanobis metric learning”. In Section 1.2 (Related work), we describe existing work on Mahalanobis metric learning in detail, and concretely mention differences between our work and existing work.
>
> If the reviewer has specific suggestions or examples of the type of description they would like to see in our paper, we would be happy to have a discussion centered around that.
>
>
> >There are inconsistencies in expression or format. For example, “metric model” and “measurement model”, “In metric learning” and “In our theoretical results”.
>
> We thank the reviewer for pointing this out. We will do a thorough edit to ensure consistency in the camera-ready version.
>
> That said, we emphasize that:
> - The model of human perception (“metric model”) and the schema under which we collect responses (“measurement model”) are two distinct things.
> - Furthermore, “In metric learning” refers broadly to the field of metric learning, whereas “in our theoretical results” refers specifically to our estimator construction and analysis.
> 	- Such expressions (or slight variations) are commonly used in the literature.
> 	- For example, the phrase “in metric learning” appears in the abstract of a well-cited survey paper [Kulis13], whereas the phrase “in our theoretical analyses” appears in a foundational paper on low-rank matrix sensing [NW11].
>
> >The usage of some symbols is inconsistent with the conventions of metric learning. For example, y often indicates a label in metric learning, there is a symbol with an arrow to indicate a constant that has not seen such usage, and there is a greater than or equal symbol with a wavy line that does not understand its meaning.
>
> We will conduct a full review to iron out any notational issues.
>
> We emphasize that:
> - We appreciate the reviewer bringing this up as a source of potential confusion and will consider changing notation for future drafts.
> - The notation $y$ was chosen due to its connection to the matrix sensing literature.
> 	- In particular, measurement values in matrix sensing literature of the form $y = \langle A, \Sigma^\star \rangle + \eta$ is the standard.
> 	- Towards this end, we clearly define y in section 2.1
> - Assuming the symbol being referred to is $\eta^\uparrow$, this is defined on line 169 as the upper bound of the noise. This is referred to again in line 246, right before Theorem 1. Furthermore, the combined quantity $y^\uparrow$ is defined in the Theorem statement and does not appear before this definition. We are happy to revise these quantities to be $\eta^{max}$ and $y^{max}$ to avoid any unwarranted confusion.
> - We follow the standard statistics literature and use $\lesssim$ and $\gtrsim$ to compare the order of two quantities (defined in Appendix B on Page 13 for completeness).
>
> > There are major problems with the format of references, such as the lack of journal names and page numbers in some citations, as well as the problem of inconsistent cases between the paper name and the journal name.
>
> > There are some grammatical errors, please proofread carefully. For example, “such as in conducting surveys, grading assignments,…”should remove “in”, inconsistent case after the initial letter of the colon.
>
> Thank you for bringing this to our attention. We will correct any reference inconsistencies and grammatical mistakes in the camera-ready version.

---

### Official Review · Reviewer_sAC6 · 2023-07-06

**Soundness:** 3 good
**Presentation:** 3 good
**Contribution:** 3 good
**Rating:** 6
**Confidence:** 3

**Summary:**

 this paper introduces the Perceptual Adjustment Query (PAQ) mechanism for collecting human feedback in a cognitively lightweight manner, which combines advantages from both ordinal and cardinal queries.
The authors apply PAQs to the problem of  metric learning under a Mahalanobis metric.



**Strengths:**

- The introduction of the Perceptual Adjustment Query (PAQ) as a hybrid of ordinal and cardinal queries is a novel contribution. This mechanism offers the advantages of both query types.

- The paper applies the PAQ scheme to the framework of metric learning for human perception. This application demonstrates the utility of PAQs in learning distance metrics and capturing similarities and dissimilarities in perceptual data.

- The authors provide sample complexity guarantees and numerical experiments for the proposed estimator, which offer insights into the understanding of the PAQ-based approach.




**Weaknesses:**


1. The paper is based on the idea that the PAQ style of queries is more useful than ordinal or cardinal queries. But it is not clear how this can be tested. The paper would benefit from discussing potential real-world applications where the PAQ mechanism could be better than other relatedapproaches. This discussion would provide insights into the impact of the proposed approach.
2. Similarly, I think the paper would benefit from exploring additional tasks other than Mahalanobis metric learning,
3. Limited Evaluation: While the paper mentions ”extensive numerical simulations” to demonstrate the performance and properties of the estimator, it would be beneficial to provide more experimental results. Additional evaluation metrics and comparisons with alternative approaches would strengthen the paper's contributions. For example:
- Apply this approach to tasks other than Mahalanobis metric learning ?
- Comparison: The experiments focus solely on evaluating the PAQ-based metric learning method. Including a comparative analysis with existing approaches or baselines would provide a more comprehensive evaluation and further validate the proposed approach.
- How is the data collected different from ordinal/cardinal queries? Can this difference be demonstrated experimentally?



**Questions:**

same as above.

**Limitations:**

yes.

---

> ### Author Rebuttal · Authors · 2023-08-10
>
> Thank you for your thoughtful comments and feedback. We are grateful that you appreciate the advantages of PAQs and our sample complexity guarantees.
>
> > Limited Evaluation: While the paper mentions ”extensive numerical simulations” to demonstrate the performance and properties of the estimator, it would be beneficial to provide more experimental results. Additional evaluation metrics and comparisons with alternative approaches would strengthen the paper's contributions.
>
> > Comparison: The experiments focus solely on evaluating the PAQ-based metric learning method. Including a comparative analysis with existing approaches or baselines would provide a more comprehensive evaluation and further validate the proposed approach.
>
> We have performed a new set of simulation experiments comparing PAQs against two often used queries in metric learning: Triplets and binary comparisons (please see the global response and attached PDF for full details). To summarize, PAQs drastically outperform these two alternative approaches, greatly reducing the number of queries needed to achieve a particular error level.
>
> The numerical simulations that we presented in the paper are exactly as noted: to verify our theoretical results. We view this paper as one that 1.) formalizes the query and 2.) establishes a theoretical foundation in an important area that highlights novel technical challenges that arise when working with a new query paradigm. We see user studies as an important avenue for future work.
>
> > The paper would benefit from discussing potential real-world applications where the PAQ mechanism could be better than other related approaches. This discussion would provide insights into the impact of the proposed approach.
>
> We agree with the reviewer that our paper would benefit from a prolonged discussion of real-world applications and plan on incorporating such a discussion in our revision. We emphasize that the PAQ is broadly applicable to many downstream tasks, and highlight two such examples for which the PAQ is well-suited in the global response: characterizing color perception and querying for housing similarity.
>
> > Similarly, I think the paper would benefit from exploring additional tasks other than Mahalanobis metric learning,
> > Apply this approach to tasks other than Mahalanobis metric learning ?
>
> We emphasize that we purposely chose to limit the scope of this work to Mahalanobis metric learning precisely due to the unique and novel technical challenges it poses from a theoretical perspective. Because metric learning can be applied to a myriad of applications (see global response for an non-exhaustive list), many works in this field are self-contained, focusing only on metric learning. However, in our revision, we will include descriptions for how to apply PAQs to different settings and tasks.
>
> Specifically, as mentioned in the global common response, the focus on metric learning allows us to bring two novel problem to the high-dimensional statistics and low-rank matrix sensing communities: the un-studied inverted measurement scheme and estimator design under simultaneously heavy-tailed and bias-inducing sensing matrices. We present, to the best of our knowledge, the first estimator that under these novel problem settings.
>
> > How is the data collected different from ordinal/cardinal queries? Can this difference be demonstrated experimentally?
>
> The user’s response to a PAQ is the item along the path where the transition from similar items to dissimilar items occurs. This response is inherently much richer than ordinal query responses, which only reveal quantized similarity judgements (e.g., “Item A is more similar to the reference item than item B”). The PAQ response item can equivalently be viewed as a scaling of the query vector $a$. We note that while the response can be viewed as a scalar, it is inherently different from a cardinal response. PAQ responses are anchored to a reference object, thus providing context for users to make similarity judgments. This context is missing from cardinal queries, which ask for an absolute similarity rating (e.g., “On a scale from 1 - 10, how similar are these items?”), leading to issues such as miscalibration.
>
> We demonstrate the effect of the different response types between PAQs and common ordinal queries in our new set of simulation results (please see global response and attached PDF).

---

> > ### Comment · Reviewer_sAC6 · 2023-08-22
> >
> > Thank you for your answer, and for the additional simulation experiments.
> > My view of the paper remains positive.

---

### Official Review · Reviewer_Ehde · 2023-07-10

**Soundness:** 3 good
**Presentation:** 3 good
**Contribution:** 3 good
**Rating:** 6
**Confidence:** 3

**Summary:**

This paper proposes to collect users' perceptual adjustment queries (PAQ) to capture change points in a semantic space. In contrast to ordinal and cardinal data, PAQ better aligns with human perception and is cheap to collect. PAQ can be used to learn the semantic space manifold by solving a row-rank optimization algorithm. The paper proposes theoretical guarantees accompanied by numerical results.

**Strengths:**

1. PAQ is conceptually novel, easy to collect, and aligns with human perception of concepts.
2. The metric learning algorithm is straightforward and is backed with theoretical guarantees.

**Weaknesses:**

1. The paper does not provide a downstream use case for PAQ.
2. Selecting the direction $a$ is challenging in high dimensions.
3. The truncation trick in line 219 seems ad hoc to me.

**Questions:**

The demonstration in Figure 2. looks very similar to embedding spaces of images. Have you considered using PAQ to learn the underlying embedding spaces for languages and images?

**Limitations:**

See "weaknesses."

---

> ### Author Rebuttal · Authors · 2023-08-10
>
> Thank you for your comments and questions. We are delighted that you find our proposed query conceptually novel, well-motivated, and easy to implement.
>
> > The paper does not provide a downstream use case for PAQ.
>
> We agree with the reviewer that a discussion of downstream use cases for PAQs would improve the paper. We emphasize that the PAQ is broadly applicable to many downstream tasks, but we keep the PAQ framework general from the perspective of query design in our presentation. In the global response, we highlight two examples of applications for which the PAQ is well-suited: querying for housing similarity and characterizing color perception.
>
> >  Selecting the direction a is challenging in high dimensions.
>
> We agree that the problem of selecting an “optimal” query direction is extremely interesting, and presents an avenue for future work. We draw $a$ from the standard $d$-dimensional normal distribution, which means that the direction of $a$ is uniformly distributed in $\mathbb{R}^d$. Due to the inverted nature of our measurements, it was not obvious that this choice of $a$ would allow us to perform consistent and theoretically justified estimation. However, we are able to design an estimator with this choice of $a$ and prove non-trivial theoretical bounds. In practice, selecting semantically meaningful directions in an active manner is an interesting line of future exploration.
>
> > The truncation trick in line 219 seems ad hoc to me.
>
> Due to the inverse quadratic term in Equation (6), the sensing matrices are heavy-tailed.
> As a result, we utilize truncation to mitigate the effects of the heavy tail to ensure our measurements do not deviate “too far” from the probabilistically expected response. A similar technique has been shown to mitigate heavy tails when the truncation threshold is set carefully [FWZ21].
>
> > The demonstration in Figure 2. looks very similar to embedding spaces of images. Have you considered using PAQ to learn the underlying embedding spaces for languages and images?
>
> We thank the reviewer for this suggestion. We consider an exploration of leveraging PAQs in learning representations an exciting and critical exploration of future work. We hope that by combining the advantages of cardinal and ordinal feedback, any representations learned from PAQ responses will inherit the strengths of representations learned solely from ordinal or cardinal feedback.
>
> We emphasize that the PAQ is broadly applicable in a number of downstream applications, two of which we highlight in the global response.
>
> [FWZ21] J. Fan, W. Wang, Z. Zhu, “A shrinkage principle for heavy-tailed data: High-dimensional robust low-rank matrix recovery,” in the Annals of Statistics 2021.

---

### Author Rebuttal · Authors · 2023-08-10

We thank the reviewers for their time and thoughtful feedback. The reviews especially recognized the novelty of the parametric adjustment query (reviewers Ehde, sAC6, eVNV, H5Pr) in combining “the advantages of both cardinal and ordinal queries” to alleviate cognitive burden on respondents while maintaining information richness.

Below, we address three common comments among the reviewers. Other comments are responded individually to the reviewer’s thread.

**Main contributions.** We emphasize three main contributions of the paper:
1.) The proposal of using parametric adjustment query (PAQ) for data collection
2.) The resulting inverted measurement scheme, and formulation of the low-rank matrix estimation problem with an unconventional construction of the sensing matrices, which arises when applying PAQs to metric learning.
3.) A novel estimator and corresponding theoretical sample complexity guarantees

In particular, we believe contributions 2.) and 3.) are of independent methodological interest, especially to the high-dimensional statistics community. The inverted measurement scheme, to our knowledge, is unstudied and poses several major technical challenges from an estimator design perspective. Namely, inverted measurements violate two key assumptions found in the standard matrix sensing measurement model: independent sensing matrices and noise, and “well-behaved” (non-heavy tailed) sensing matrices. Violations of these two assumptions result specifically in estimator bias. We take advantage of the specific properties of the estimation bias, namely that it scales with noise variance, to employ averaging in a way to reduce bias. In all, our estimator is designed specifically to address these technical challenges, and is of independent interest to the low-rank matrix sensing community.

**Comparison against other queries (Reviewers sAC6, eVNV, H5Pr).** We performed a new set of simulation experiments comparing PAQs against two other common query types in the case of learning a low-dimensional Mahalanobis metric. We compare against triplets (“given reference item A, which of items B and C are most similar to A?”) and binary comparisons (“Are items A and B similar?”) in both the noisy and noiseless settings, reporting normalized estimation error, and attach the associated results in our response PDF. We will update our camera-ready version with these experiments.

To summarize the results: In both the noiseless and noisy regimes, PAQ outperforms both common baselines substantially. For example, in the noisy regime with 800 responses, the normalized estimation error is
- PAQ: 0.18
- Triplets: 0.73
- Binary comparison: 0.96

**This is a drastic reduction in the number of samples needed to achieve good estimation error, demonstrating the power of learning from PAQ responses over established queries.**

**Downstream application of PAQs (Reviewers Ehde, sAC6, H5Pr).** We keep the PAQ framework general from the perspective of query design. We showcase the utility of PAQs on the widely-used metric learning problem, which has several applications, such as similarity learning, preference learning [XD20, CMVN22], classification [WS09, KHWRB12], and representation learning [JJN16]. When applied to metric learning, it can be incorporated with applications such as learning image embeddings.

One example of a use case of PAQs for similarity learning is specifically mentioned by reviewer H5Pr: housing price similarity. As the user moves along the path, they see gradually changing features such as housing price, square footage, monthly fees and taxes, estimated repair costs, etc, to which they mark the first instance where they feel the house is no longer “worth it”.

Aside from metric learning, PAQ can be used to elicit data for other human perception tasks. For example, one can study color perception to characterize color-blindness. For a user with red-green color blindness, we can present the user with an image of a red square and a sequence of points that slowly transitions from red to green, and ask them to drag the slider until they perceive a difference in colors. The answers to a set of PAQs would allow us to characterize how the user perceives colors and help tailor color correction for that user. We briefly mentioned this application along with others in Sec 6, and will include the above extended discussions in the longer camera-ready version. For example, for a user with red-green color blindness, we can present the user with an image of a red square and a sequence of colors that slowly transitions from red to green, and ask them to drag the slider until they perceive a difference in colors. In such a setting PAQs thrive by presenting users both with context (the full sequence of colors and the reference color) while eliciting a response that cuts to the core of what we want to learn: At what point can you start distinguishing colors? In this scenario, asking a user for ordinal feedback such as binary comparisons quantizes differences in color perception to the point where finding this transition point is difficult. On the other hand, asking users for a cardinal rating of color similarity is an extremely ill-defined task prone to the typical drawbacks of cardinal data elicitation, such as miscalibration and scale drift.

[XD20] A. Xu and M. A. Davenport, “Simultaneous Preference and Metric Learning from Paired Comparisons,” in NeurIPS 2020
[CMVN22] G. Canal, B. Mason, R. K. Vinayak, R. Nowak, “One for All: Simultaneous Metric and Preference Learning over Multiple Users,” in NeurIPS 2022
[WS09] K. Weinberger, L. K. Saul, “Distance metric learning for large margin nearest neighbor classification,” in JMLR 2009
[KHWRB12] M. Kostinger, M. Hirzer, P. Wohlhart, P. M. Roth, H. Bischof, “Large Scale Metric Learning from Equivalence Constraints,” in CVPR 2012
[JJN16] L. Jain, K. G. Jamieson, R. Nowak, “Finite sample prediction and recovery bounds for ordinal embedding,” in NeurIPS 2016

---

### Author Response · Authors · 2023-08-16

We want to thank the reviewers again for the time and effort of their reviews. We are happy to address any remaining questions that reviewers have before the discussion period ends on August 21.

We also wish to inform reviewers that, at the suggestion of reviewer H5Pr, we have conducted an additional set of simulations comparing the performance of PAQs against tuplewise ranking queries [1]. As the authors of [1] mention, to perform inference with these tuplewise ranking queries, one decomposes the ranking query response into a set of triplet responses. Thus, informativeness of ranking queries is relatively limited in comparison to PAQs, in that responses quantize statements of human perception. Furthermore, as the size of the set of items to be ranked grows, the cognitive burden on the user increases dramatically [2], [3]

Plots of relative error vs. number of queries look similar to the already attached figures but we are not allowed to attach updated figures; instead, please see the table below for our results. We denote ranking-$k$ to mean $k$ items are ranked in relation to a reference object, meaning $k+1$ total items are presented to a user, and report normalized estimation error.

| Number of responses | 500       | 1000      | 1500      | 2000      | 2500      |
|---------------------|-----------|-----------|-----------|-----------|-----------|
| PAQ                 | **0.396** | **0.135** | **0.094** | **0.073** | **0.063** |
| ranking-$3$         | 0.621     | 0.398     | 0.309     | 0.253     | 0.225     |
| ranking-$4$         | 0.423     | 0.268     | 0.210     | 0.178     | 0.156     |

In summary, PAQs still outperform the more complex tuplewise ranking queries whenever a reasonably small error is desired, i.e., to achieve reasonable estimation error, one needs substantially more ranking query responses than PAQ responses. For example, to achieve normalized estimation error of 0.2 in the noisy regime, one needs
- PAQ: Between 700 - 800 responses
- ranking-$3$: Greater than 2500 responses
- ranking-$4$: Between 1500 - 2000 responses
- Triplets: Greater than 2500 responses

---
[1] G. Canal, S. Fenu, C. Rozell. ​​”Active Ordinal Querying for Tuplewise Similarity Learning,” in AAAI 2020

[2] H. A. Simon, “A behavioral model of rational choice,” The quarterly journal of economics, vol. 69, no. 1, pp. 99–118, 1955.

[3] G. Miller, “The magical number seven, plus or minus two: Some limits on our capacity for processing information.” Psych. Rev., vol. 63, no. 2, p. 81, 1956.

---

### Decision · Program_Chairs · 2023-09-21

**Decision:**

Accept (poster)

**Comment:**

This paper introduced the Perceptual Adjustment Query (PAQ) as a hybrid of ordinal and cardinal queries which is well motivated by practical problems.  The authors propose two techniques including sample averaging and truncation to overcome challenges that do not present in classical low-rank matrix estimation problems and prove statistical bounds on the estimator error.

The reviews collectively acknowledge the innovation of PAQ and the proposed approach.  There is also a nice theory about its statistical error estimation.   While certain reviewers expressed reservations regarding the limited experimentation and requested additional empirical evidence, the authors' response is notably convincing to me.

Given these factors, I am inclined to recommend the paper for acceptance due to its inventive approach, solid theoretical underpinning, and the author's adept addressing of reviewer concerns.